# Profiling of myristoylation in *Toxoplasma gondii* reveals an *N*-myristoylated protein important for host cell penetration

Malgorzata Broncel[1], Caia Dominicus[1], Luis Vigetti[2], Stephanie D Nofal[1], Edward J Bartlett[3], Bastien Touquet[2], Alex Hunt[1], Bethan A Wallbank[1], Stefania Federico[4], Stephen Matthews[5], Joanna C Young[1], Edward W Tate[3], Isabelle Tardieux[2], Moritz Treeck[1]*

[1]Signalling in Apicomplexan Parasites Laboratory, The Francis Crick Institute, London, United Kingdom; [2]Institute for Advanced Biosciences, Team Membrane Dynamics of Parasite-Host Cell Interactions, CNRS UMR5309, INSERM U1209, Université Grenoble Alpes, Grenoble, France; [3]Department of Chemistry, Imperial College London, Molecular Sciences Research Hub, White City Campus, London, United Kingdom; [4]The Peptide Chemistry STP, The Francis Crick Institute, London, United Kingdom; [5]Department of Life Sciences, Imperial College London, South Kensington, London, United Kingdom

**Abstract** *N*-myristoylation is a ubiquitous class of protein lipidation across eukaryotes and *N*-myristoyl transferase (NMT) has been proposed as an attractive drug target in several pathogens. Myristoylation often primes for subsequent palmitoylation and stable membrane attachment, however, growing evidence suggests additional regulatory roles for myristoylation on proteins. Here we describe the myristoylated proteome of *Toxoplasma gondii* using chemoproteomic methods and show that a small-molecule NMT inhibitor developed against related *Plasmodium spp.* is also functional in *Toxoplasma*. We identify myristoylation on a transmembrane protein, the microneme protein 7 (MIC7), which enters the secretory pathway in an unconventional fashion with the myristoylated N-terminus facing the lumen of the micronemes. MIC7 and its myristoylation play a crucial role in the initial steps of invasion, likely during the interaction with and penetration of the host cell. Myristoylation of secreted eukaryotic proteins represents a substantial expansion of the functional repertoire of this co-translational modification.

*For correspondence:
moritz.treeck@crick.ac.uk

## Introduction

Toxoplasmosis currently affects approximately one third of the world's population (*Robert-Gangneux and Dardé, 2012*). It is caused by the obligate protozoan parasite *Toxoplasma gondii* originating from the phylum Apicomplexa. While the majority of human infections are asymptomatic, the disease manifests its severity in immunocompromised individuals, such as those receiving chemotherapy and transplants or in HIV/AIDS patients (*Montoya and Liesenfeld, 2004*). Key steps in the successful propagation of *Toxoplasma* infection in the acute phase are orchestrated cycles of invasion and egress of tachyzoites from host cells (*Black and Boothroyd, 2000*). These crucial processes are regulated by several post-translational modifications (PTMs), such as phosphorylation (*Gaji et al., 2015*; *Jacot and Soldati-Favre, 2012*; *Lourido et al., 2010*; *Treeck et al., 2014*), ubiquitination (*Silmon de Monerri et al., 2015*), and also protein lipidation, such as palmitoylation and myristoylation (*Alonso et al., 2012*; *Frénal et al., 2014*).

While the extent of protein palmitoylation in *Toxoplasma* has been investigated (*Caballero et al., 2016*; *Foe et al., 2015*), the myristoylated proteome remains largely uncharacterised. *N*-

**eLife digest** A microscopic parasite known as *Toxoplasma gondii* infects around 30% of the human population. Most infections remain asymptomatic, but in people with a compromised immune system, developing fetuses and people infected with particular virulent strains of the parasite, infection can be fatal. *T. gondii* is closely related to other parasites that also infect humans, including the one that causes malaria. These parasites have complex lifecycles that involve successive rounds of invading the cells of their hosts, growing and then exiting these cells. Signaling proteins found at specific locations within parasite cells regulate the ability of the parasites to interact with and invade host cells. Sometimes these signaling proteins are attached to membranes using lipid anchors, for example through a molecule called myristic acid.

An enzyme called NMT can attach myristic acid to one end of its target proteins. The myristic acid tag can influence the ability of target proteins to bind to other proteins, or to membranes. Previous studies have found that drugs that inhibit the NMT enzyme prevent the malaria parasite from successfully invading and growing inside host cells. The NMT enzyme from *T. gondii* is very similar to that of the malaria parasite. Broncel et al. have shown that the drug developed against *P. falciparum* also inhibits the ability of *T. gondii* to grow. These findings suggest that drugs against the NMT enzyme may be useful to treat diseases caused by *T. gondii* and other closely-related parasites.

Broncel et al. also identified 65 proteins in *T. gondii* that contain a myristic acid tag using an approach called proteomics. One of the unexpected 'myristoylated' proteins identified in the experiments is known as MIC7. This protein was found to be transported onto the surface of *T. gondii* parasites and is required in its myristoylated form for the parasite to successfully invade host cells. This was surprising as myristoylated proteins are generally thought to not enter the pathway that brings proteins to the outside of cell.

These findings suggest that myristic acid on proteins that are secreted can facilitate interactions between cells, maybe by inserting the myristic acid into the cell membrane.

myristoylation is an irreversible, predominantly co-translational covalent addition of myristic acid to an N-terminal glycine (*Boutin, 1997*; *Gordon et al., 1991*). Functionally, myristoylation often primes proteins for subsequent palmitoylation and a stable protein-membrane association (*Martin et al., 2011*; *Wright et al., 2010*), however, it has also been shown to facilitate protein-protein interactions (PPIs) (*Chow et al., 1987*; *Mousnier et al., 2018*), affect protein activity (*Zhu et al., 2019*) as well as structure and stability (*Zheng et al., 1993*). It is catalysed by *N*-myristoyl transferase (NMT), which is conserved across many organisms, including *Toxoplasma*, and has been reported to be a prominent drug target in fungal (*Devadas et al., 1995*; *Nagarajan et al., 1997*), *Trypanosome* (*Frearson et al., 2010*; *Wright et al., 2016*) and *Leishmania* infections (*Hutton et al., 2014*; *Wright et al., 2015*). In *Plasmodium falciparum* (the major causative agent of malaria), inhibition of NMT leads to severe pleiotropic consequences affecting parasite development (*Schlott et al., 2019*; *Wright et al., 2014*), highlighting the importance of myristoylation for parasite survival and progression.

An N-terminal glycine (MG motif) is a requirement, but not a predictor of myristoylation. Approximately 6% of all gene products in *Toxoplasma* contain an N-terminal glycine and an in silico prediction of myristoylation suggests that ~ 1.8% of all *T. gondii* gene products are modified (*Alonso et al., 2019*). The functional significance of myristoylation has been described for only a few *T. gondii* proteins, and mainly in conjunction with adjacent palmitoylation that promotes stable membrane attachment. These proteins include key signal mediators in parasite egress and invasion, for example CDPK3 (*Garrison et al., 2012*; *McCoy et al., 2012*), PKG (*Brown et al., 2017*), PKAr (*Jia et al., 2017b*; *Uboldi et al., 2018*); proteins involved in invasion, for example IMP1 (*Jia et al., 2017a*); parasite gliding, for example GAP45 and GAP70 (*Frénal et al., 2010*); division, for example F-box protein 1 and ISP1, 2, 3 (*Baptista et al., 2019*; *Beck et al., 2010*); and correct rhoptry positioning required for invasion, for example ARO (*Cabrera et al., 2012*; *Mueller et al., 2013*). Collectively, these studies show key roles for myristoylation throughout the parasite's lytic cycle, but the function of myristoylation in the absence of palmitoylation and its relationship to other PTMs remains poorly described.

By combining several chemoproteomic tools for substrate identification with a small-molecule NMT inhibitor, we provide experimentally-validated libraries of myristoylated as well as glycosyl-phosphatidylinositol (GPI) anchored proteins in *T. gondii*. We identify all the previously reported myristoylated proteins, as well as novel substrates with heterogeneous localisations and variable functions across the lytic cycle. Furthermore, by analysing substrate orthology in other Apicomplexans we provide new clues to the identity of previously uncharacterized myristoylated proteomes across the phylum. We validate the presence and elucidate the functional importance of myristoylation for the microneme protein MIC7, a predicted type I transmembrane protein. Utilizing conditional substrate depletion and complementation with wild-type (cWT) and myristoylation mutant (cMut) versions, we demonstrate that myristoylation of MIC7 is functionally important in host cell invasion. Taken together, our study identifies a large proportion of the *Toxoplasma* myristoylated proteome and points to unexpected and novel functions of myristoylation in *Toxoplasma* that extend beyond priming for palmitoylation and stable membrane attachment.

## Results

### Metabolic labelling allows for enrichment and visualisation of myristoylated and GPI-anchored proteins in *Toxoplasma*

To visualise the extent of myristoylation in *Toxoplasma,* we adapted a metabolic labelling approach that has previously been applied to mammalian cells (**Broncel et al., 2015**; **Thinon et al., 2014**) and protozoan parasites (**Wright et al., 2014**; **Wright et al., 2016**; **Wright et al., 2015**). In this workflow, a myristic acid (Myr) analogue containing a terminal alkyne group (YnMyr) is added to cell culture upon infection with *Toxoplasma* tachyzoites (**Figure 1A**). The hydrophobic nature of YnMyr allows for optimal biomimicry of myristate and conversion into the active co-substrate YnMyr-CoA in situ, while the alkyne tag allows for NMT-mediated metabolic labelling of both host and parasite substrate proteins. Upon cell lysis, labelled proteins are liberated and conjugated to azide-bearing multifunctional capture reagents by a click reaction (**Heal et al., 2012**). The conjugation process introduces secondary labels, like biotin and fluorophores, allowing for substrate enrichment on streptavidin beads and visualisation via in-gel fluorescence (igFL), respectively. To investigate the extent of YnMyr incorporation, intracellular tachyzoites were treated with either Myr or increasing concentrations of YnMyr for 16 hr. Within this timeframe protein labelling in vivo did not appear to exert any toxic effects on *Toxoplasma* parasites. Labelled proteins were then conjugated to a capture reagent and resolved by SDS-PAGE. As visualised by igFL, the labelling was concentration-dependent with only negligible background (**Figure 1—figure supplement 1A**). In addition, the extent of labelling did not seem to depend on parasite localisation inside or outside the host cell, and was efficiently out-competed by excess myristate, indicating that YnMyr is an effective mimic of myristate in *Toxoplasma* parasites (**Figure 1—figure supplement 1B**). To estimate the efficiency of substrate enrichment, we took advantage of the biotin moiety that enables a streptavidin-based pull down. Using igFL as readout we observed robust enrichment of protein substrates in a YnMyr-dependent manner, and detected very little background in controls (**Figure 1—figure supplement 1C**).

It has been reported that in *Plasmodium* parasites, YnMyr can be incorporated not only at N-terminal glycines via amide bonds, but also through ester-linked incorporation of myristate into GPI anchors (**Wright et al., 2014**). These two distinct types of labelling can be readily distinguished by their different sensitivity to base treatment; amide bonds are stable in basic conditions, whereas ester bonds are hydrolysed. To visualise the extent of YnMyr incorporation into GPI anchors in *Toxoplasma*, we performed base treatment prior to enrichment of substrate proteins and observed a reduction of igFL signal for selected enriched bands (**Figure 1B**). To further validate the base treatment approach, we probed known *N*-myristoylated and GPI-anchored *Toxoplasma* proteins, GAP45 and SAG1, for their ability to be enriched in a base-dependent manner. In the absence of treatment, both proteins were robustly pulled down with YnMyr, while upon base treatment, only GAP45 remained enriched, confirming that it is a true myristoylation substrate (**Figure 1C**). Collectively, we confirmed that YnMyr is a robust and high-fidelity myristate analogue and demonstrated that it can be applied to profile both *N*-myristoylated and GPI-anchored proteins in live *T. gondii*.

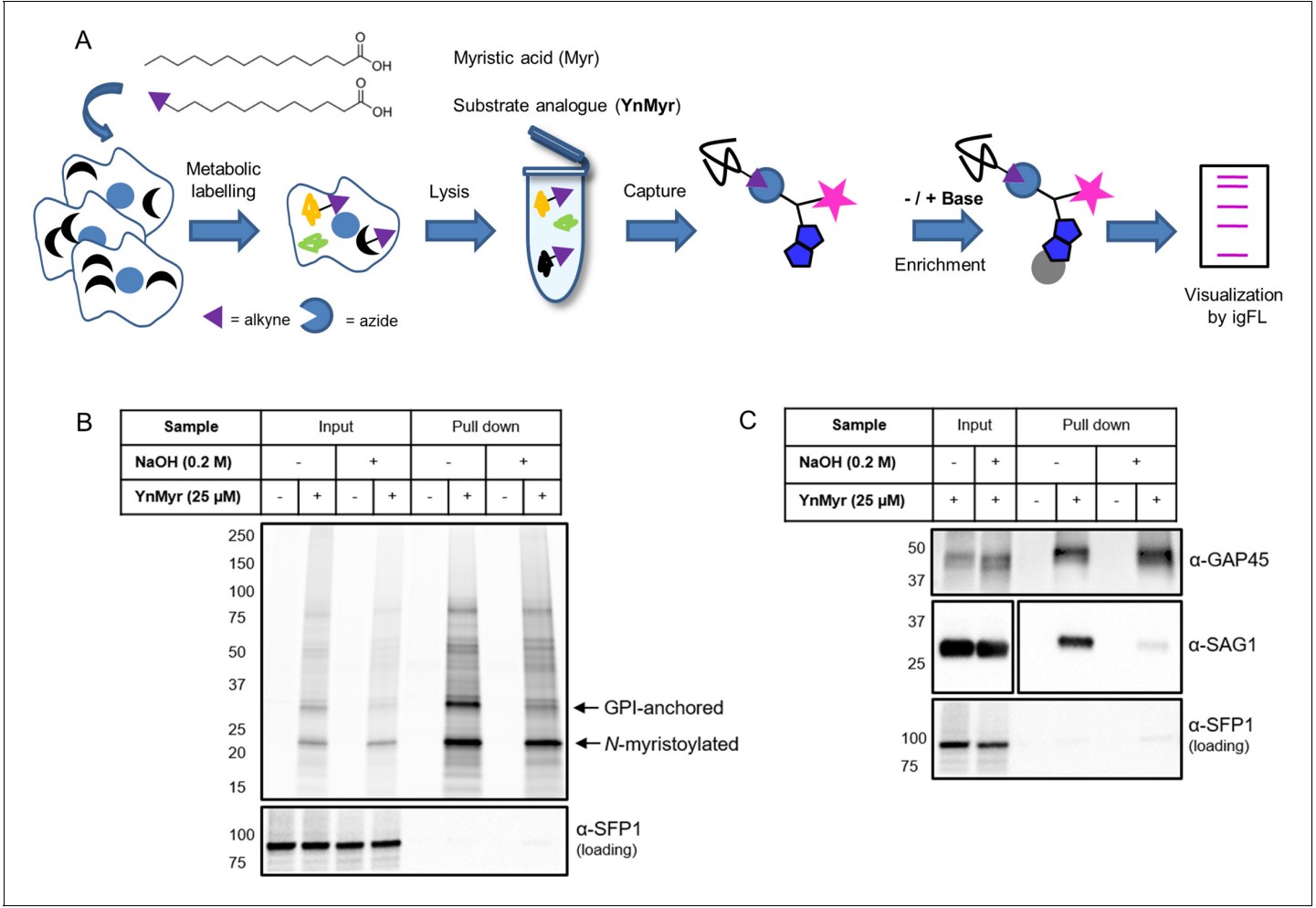

**Figure 1.** Metabolic labelling allows for enrichment and visualisation of myristoylated and GPI-anchored proteins in *T. gondii.* (**A**) Metabolic labelling workflow. (**B**) In gel fluorescence visualisation of YnMyr-dependent enrichment without and with the base treatment (top) and western blot with α-SFP1 (TGGT1_289540) showing the loading control (bottom). (**C**) Western blot analysis of YnMyr-dependent pull down for known myristoylated and GPI-anchored proteins GAP45 and SAG1, respectively. White space was used to indicate where gel lanes were not contiguous. See also *Figure 1—figure supplement 1*.

The online version of this article includes the following figure supplement(s) for figure 1:

**Figure supplement 1.** Metabolic labelling optimisation.

## Proteomic identification of YnMyr-enriched proteins in *T. gondii*

To confidently identify YnMyr-labelled proteins in *Toxoplasma,* we applied state-of-the-art mass spectrometry (MS)-based proteomics combined with validated chemical tools (*Figure 2—figure supplement 1A*); (*Broncel et al., 2015*; *Speers and Cravatt, 2005*; *Thinon et al., 2014*; *Wright et al., 2014*). We started with a small-scale pilot experiment to test our workflow and differentiate between *N*-myristoylation-based enrichment and GPI-anchored substrates. We metabolically labelled intracellular tachyzoites of the RH strain (*Huynh and Carruthers, 2009*) with either YnMyr or Myr each at 25 μM for 16 hr. We then lysed the infected cell monolayers and performed the click reaction with the azido biotin capture reagent (reagent **1**) to facilitate YnMyr-dependent enrichment of labelled proteins. To distinguish myristoylated from GPI-anchored substrates, we applied base treatment prior to the streptavidin-based pull down. Following trypsin digestion, we analysed samples by LC-MS/MS and performed label free quantification (LFQ) of enriched proteins. We quantified 2363 human and *Toxoplasma* proteins, 349 of which were parasite proteins with YnMyr intensities irrespective of base treatment (*Supplementary file 1*). To identify GPI-anchored proteins, we calculated $\log_2$ fold changes between base-treated and untreated samples. To threshold we utilised the

least extreme negative value (log$_2$ fold change < −1) quantified from all Surface Antigen Proteins (SAGs) detected in our study, which are known to be GPI-anchored (*Figure 2—figure supplement 1B*). This selection strategy yielded 52 substrates, that included known and predicted GPI-anchored proteins (*Supplementary file 1*). To identify myristoylated proteins we utilised a stringent selection method based on three criteria: a) robust YnMyr/Myr enrichment (log$_2$ fold change > 2) with threshold selected based on known myristoylated proteins (*Figure 2—figure supplement 1C*), b) the presence of an MG motif and c) insensitivity to base treatment. 56 proteins met these criteria, including those previously reported as myristoylated (*Supplementary file 1*). Analysis of post enrichment supernatants did not reveal any substantial changes between proteomes of the YnMyr- and Myr-treated samples, confirming that the observed enrichment is not due to globally altered protein abundance (*Figure 2—figure supplement 1D* and *Supplementary file 1*).

After successful testing of the metabolic labelling workflow, we performed a more elaborate MS experiment using cleavable capture reagents bearing either trypsin (reagent **2**) or TEV (reagent **3**) cleavable linkers (*Figure 2—figure supplement 1A*). In contrast to a non-cleavable reagent (e.g. reagent **1**) that provides only indirect proof of substrate myristoylation, cleavable reagents allow for detection of myristoylated peptides in addition to peptides that originate from the enriched proteins (*Figure 2A*). This additional layer of confidence in MS-based substrate identification is especially important given the high level of non-myristoylation dependent background reported for metabolic labelling with YnMyr (*Broncel et al., 2015*; *Wright et al., 2016*; *Wright et al., 2015*). While reagent **2** has been validated as a tool for myristoylated protein and peptide discovery (*Broncel et al., 2015*), reagent **3** (*Speers and Cravatt, 2005*), which is expected to produce less background and improve myristoylated peptide discovery, has not previously been applied to study protein myristoylation. We therefore first tested reagent **3** in terms of YnMyr-dependent protein enrichment and observed robust pull down of potential NMT substrates (*Figure 2—figure supplement 1E*). We next generated samples for the MS workflow as described above but, instead of conjugating reagent **1,** we conjugated either **2** or **3**, each in biological triplicate, to labelled proteins via click reaction to enable myristoylation-dependent pull down. As depicted in *Figure 2A*, reagent **2** requires only a single trypsin digestion step to liberate both unmodified and myristoylated peptides in one pool. By contrast, reagent **3** requires both trypsin and TEV protease digestion and, depending on the enzyme combination, releases unmodified and myristoylated peptides in either one (TEV I) or two (TEV II) separate fractions (*Figure 2A*). In the TEV I strategy TEV protease is used to cleave proteins from beads followed by trypsin digestion of proteins into peptides. The cleavage will only occur for proteins bound via the TEV linker and not for the non-specifically bound ones, which should significantly reduce background. In the TEV II strategy, trypsin is used first to remove most proteins from the beads, only retaining the captured myristoylated peptides. These are then specifically released using TEV protease cleavage resulting in much reduced sample complexity, which increases the myristoylated peptide discovery by MS. Following digestion, all samples were subjected to LC-MS/MS, and LFQ was performed to identify proteins robustly enriched in YnMyr-dependent manner. This yielded 206 human and 117 *T. gondii* proteins bearing an N-terminal MG motif (*Supplementary file 2*). Within the parasite protein pool, we obtained statistically significant (FDR 1%, log$_2$ fold change >2) enrichment in YnMyr over Myr controls for 72 potential substrates using reagent **2** (*Supplementary file 2*). For reagent **3,** which was used in two different scenarios (TEV I and TEV II) resulting in larger variability between replicates, we utilised a fold change based threshold (log$_2$ fold change >2) and obtained 48 robustly enriched proteins (*Supplementary file 2*). Reassuringly, we observed a ~ 5 and~8 fold reduction in background in TEV I vs TEV II and TEV I vs reagent **2**, respectively, as shown by the number of proteins quantified in Myr controls (*Supplementary file 2*). Collectively we identified 76 significantly YnMyr-enriched proteins utilizing reagents **2** and **3** with an overlap of 60% (*Figure 2B*, *Supplementary file 2*) which provides substantial confidence to the accuracy of our results. Application of the same selection criteria to 206 human MG proteins identified in our study yielded 102 potential substrates. 84 of these proteins have previously been reported as myristoylated (*Broncel et al., 2015*; *Castrec et al., 2018*; *Thinon et al., 2014*), which further strengthens our substrate identification strategy.

We next focused on the identification of myristoylated peptides in samples processed with reagents **2** and **3**. Using stringent criteria for the unbiased identification of the myristoylation adduct, as well as manual validation of the acquired MS/MS spectra, we identified 31 myristoylated peptides (*Supplementary file 2*), 24 of which were detected using reagent **2**, and 20 using reagent **3**

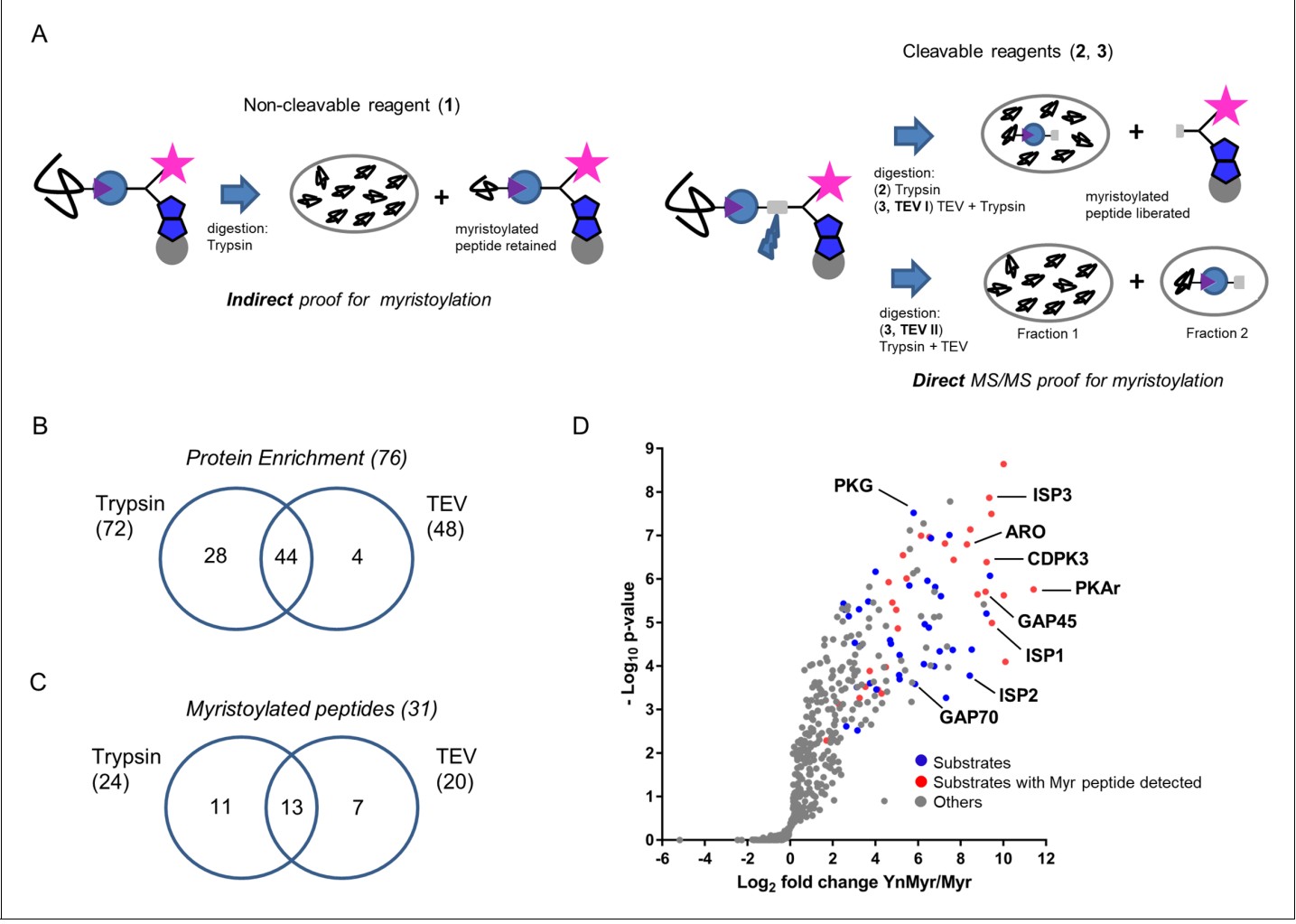

**Figure 2.** Identification of the YnMyr-enriched proteome in *T. gondii*. (**A**) Schematic representation of the MS workflow using non-cleavable and cleavable capture reagents. (**B**) Venn diagram illustrating the overlap between significantly YnMyr-enriched proteins identified with capture reagents **2** and **3**. The number of significantly enriched proteins per reagent and in total is given in parenthesis. (**C**) Venn diagram showing the overlap in myristoylated peptide discovery between the two cleavable capture reagents used in this study. The number of modified peptides identified with each reagent and in total is given in parenthesis. (**D**) Label free quantification of the log$_2$ fold changes in YnMyr enrichment over the Myr control plotted against the statistical significance for all parasite proteins detected in this study using reagent **2**. Proteins with N-terminal glycine and significant, base-insensitive enrichment with at least two capture reagents are highlighted in blue and red subject to the presence of a myristoylated peptide. All other identified proteins (YnMyr-enriched with only one reagent, background and GPI-anchors) are represented in grey. See also *Figure 2—figure supplement 1*, *Supplementary file 1* and *Supplementary file 2*.

The online version of this article includes the following figure supplement(s) for figure 2:

**Figure supplement 1.** Identification of the YnMyr-enriched proteome in *T. gondii*.

(*Figure 2C*). None of these peptides were detected in Myr controls, and the myristoylation adduct was not identified on cysteine residues. Despite almost equal numbers of peptides detected by the two reagents, the overlap was only 40% (*Figure 2C*), confirming the added value of orthogonal methods for modified peptide detection. As envisioned in our design strategy, we obtained an increase in myristoylated peptide discovery in TEV II (17) vs TEV I (12) workflow (*Figure 2—figure supplement 1F*).

Finally, to summarise our global proteomic study, we combined our results on both protein enrichment and the modified peptide levels. We filtered for proteins identified with at least two of three capture reagents or proteins for which we detected a lipid modified peptide. This resulted in

65 proteins, of which 48% have direct MS/MS evidence for protein myristoylation (*Supplementary file 2*, *Figure 2D*).

## Chemical inhibition of *Tg*NMT

Given that our global proteomic screen provided direct proof for substrate myristoylation for approximately 50% of selected proteins, we sought for an alternative strategy for substrate validation using NMT inhibitors (NMTi). Here, parasites are treated with NMTi to specifically reduce the incorporation of YnMyr into nascent proteins, which can be quantified by MS (*Thinon et al., 2014*; *Wright et al., 2014*; *Wright et al., 2016*; *Wright et al., 2015*). In the absence of a dedicated *Tg*NMTi, we used IMP-1002, a compound recently shown to inhibit NMT of *Plasmodium falciparum* (*Schlott et al., 2019*) which is related to *Toxoplasma*. Homology modelling (SWISS-MODEL, [*Waterhouse et al., 2018*]) of the *Tg*NMT sequence onto the available *Plasmodium vivax* (another important malaria causing *Plasmodium spp.)* NMT crystal structure with bound IMP-1002 (PDB: 6MB1, [*Schlott et al., 2019*]) revealed high sequence identity (57%) and showed that all residues directly involved in compound binding are conserved within the *Tg*NMT active site and therefore predicted to adopt an identical structural arrangement (*Figure 3A*). We therefore reasoned that IMP-1002 should also inhibit *Tg*NMT. To test this, we co-treated intracellular parasites 16 hr post invasion with YnMyr and increasing concentrations of the inhibitor for 5 hr and analysed the effects on YnMyr labelling of *Toxoplasma* proteins. A dose-dependent drop in igFL labelling of most protein bands was observed and further confirmed by specifically probing for CDPK1, a substrate identified herein (*Figure 3B*). Consistent with *Tg*NMT inhibition, CDPK1 pull down was reduced with increasing inhibitor concentrations. This was not a general reduction of protein levels as shown by anti-*Toxoplasma* antibodies. Plaque assays in the presence of inhibitor showed dose-dependent killing of parasites, suggesting that treatment with IMP-1002 has severe consequences for the in vitro expansion of the tachyzoite population but not for the host cells which appeared unaffected (*Figure 3C*).

Having confirmed target engagement with IMP-1002, we next performed a large-scale MS-based inhibitor response analysis. Intracellular parasites were fed with 25 µM YnMyr alone or co-incubated with 0.05 µM and 0.5 µM NMTi in biological triplicates. Samples were then clicked, pulled down and the level of protein myristoylation in response to IMP-1002 quantified by MS and LFQ. No major effect was observed for the lower concentration of the inhibitor, therefore we performed statistical analysis between triplicate samples treated with either only YnMyr, or YnMyr + 0.5 µM NMTi. We identified a statistically significant (FDR 5%, $\log_2$ fold change > 0.5) response for 56 proteins (*Figure 3D*, *Supplementary file 3*). Analysis of total proteomes from inhibited samples confirmed that the observed substrate response was not due to the altered protein abundance (*Figure 3—figure supplement 1*, *Supplementary file 3*). 49 significant responders contained the MG motif with 47 of these being significantly enriched in our previous experiment, while two proteins were not quantified. Specific dose-responses were plotted for selected proteins identified previously as significantly YnMyr-enriched (*Figure 3E*). While most showed a robust response to the highest concentration of inhibitor, CDPK3 and PKG did not, despite substantial literature evidence for myristoylation (*Brown et al., 2017*; *Garrison et al., 2012*; *McCoy et al., 2012*), including the presence of a myristoylated peptide in case of CDPK3 (*Supplementary file 2*). In fact, a total of 7 proteins for which the myristoylated peptide was detected did not respond robustly to NMT inhibition (*Supplementary file 3*). This behaviour is surprising, however a similar observation has been described for other organisms (*Thinon et al., 2014*; *Wright et al., 2014*; *Wright et al., 2016*; *Wright et al., 2015*) and could be due to low protein turnover, higher affinity for NMT, or potential interference from other modifications, such as protein *S*-acylation for example. A further seven proteins for which we obtained myristoylated peptides were not quantified at the highest concentration of IMP-1002 (*Supplementary file 3*), suggesting that their myristoylation state, and therefore enrichment, is most affected by NMT inhibition.

A significant response to NMTi was also observed for seven proteins that did not contain the MG motif (*Figure 3D*, *Supplementary file 3*). This could be due to post-translational myristoylation where proteolysis results in formation of N-terminal glycine (*Martin et al., 2011*; *Thinon et al., 2014*), a tight association in complex with an NMT substrate (*Thinon et al., 2014*), potential protein mis-annotation or an off-target effect of the inhibitor which was originally designed for *Pf*NMT. Importantly, all non MG proteins previously assigned as sensitive to base treatment, including all

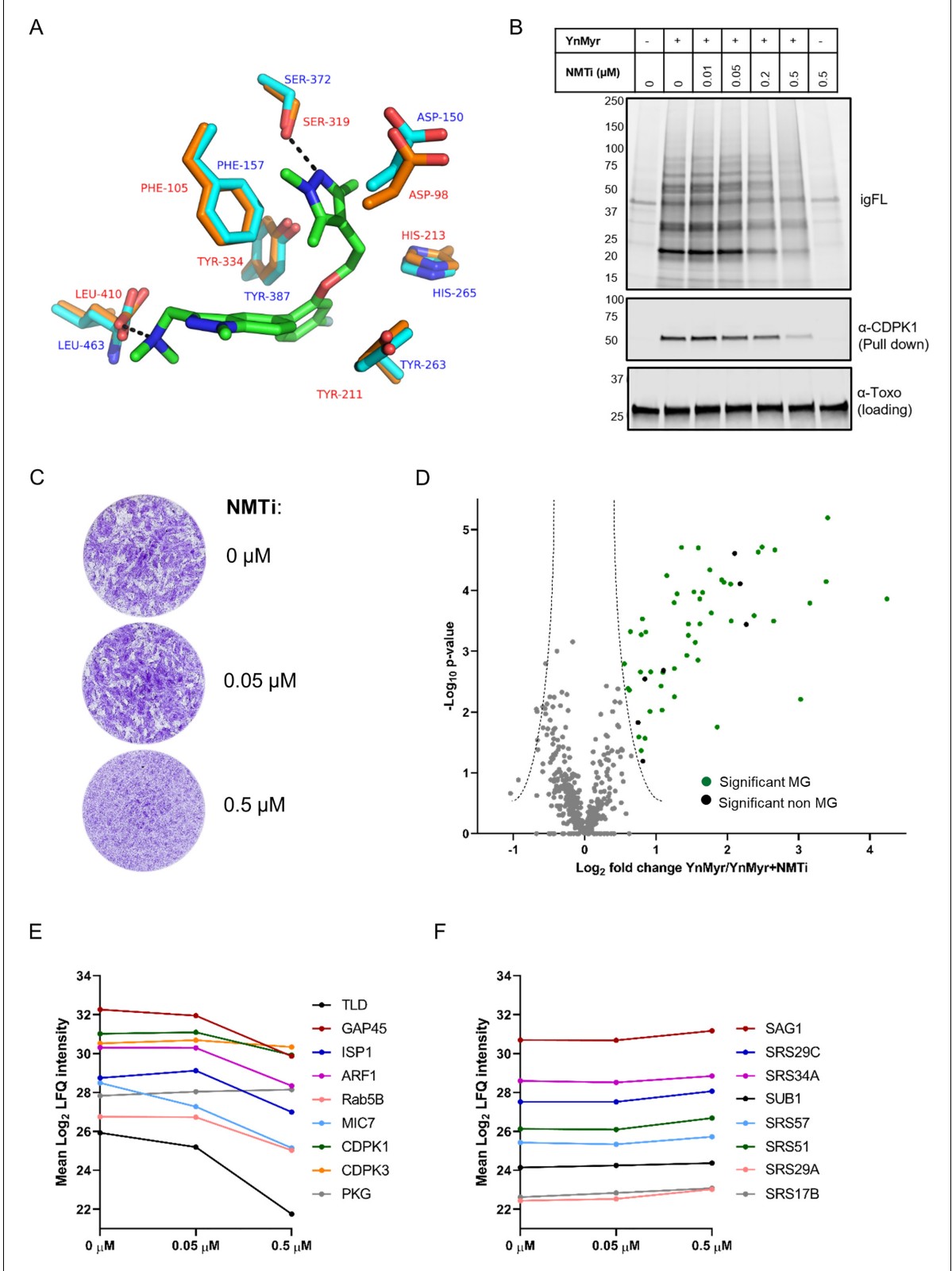

**Figure 3.** Chemical inhibition of *Tg*NMT and substrate response. (**A**) Prediction of IMP-1002 interaction with *Tg*NMT based on the *Pv*NMT crystal structure. Crystal structure of the *Pv*NMT (PDB: 6MB1, *Schlott et al., 2019*) active site (orange, red text) with IMP-1002 inhibitor bound, overlaid with a *Tg*NMT model (cyan, blue text). Hydrogen bonds between key *Pv*NMT residues (Serine and Leucine) and IMP-1002 are shown as black dashes. (**B**) Analysis of the dose response to IMP-1002 via igFL and by western blotting with CDPK1 – a substrate identified in this study. (**C**) Plaque assay

*Figure 3 continued on next page*

*Figure 3 continued*

illustrating differential killing of parasites and the host cells. The assay was performed for 5 days in three biological replicates, each in technical triplicate, representative images are shown. (D) Label free quantification of the YnMyr label incorporation into proteins in the presence of NMTi (0.5 μM) plotted against the statistical significance for all parasite proteins detected in this study. Proteins with a significant response and N-terminal glycine (MG) are highlighted in green, those without MG highlighted in black. (E) Dose response to NMTi plotted for selected proteins with significant YnMyr enrichment. (F) Dose response to NMTi plotted for selected SAG proteins assigned previously as base sensitive. See also *Figure 3—figure supplement 1* and *Supplementary file 3*.

The online version of this article includes the following figure supplement(s) for figure 3:

**Figure supplement 1.** Chemical inhibition of *Tg*NMT.

SAG proteins, showed no significant response to inhibitor (*Supplementary file 3*, *Figure 3F*) thus validating base sensitivity as a means to distinguish YnMyr incorporation at N-terminal glycines or GPI anchors.

Finally, despite no apparent effect on host cells in plaque assays (*Figure 3C*), we also observed a certain level of response to IMP-1002 for host proteins (*Supplementary file 3*). This suggests that human NMT is also targeted by this compound, however, without visible impact on the integrity of the monolayer of host cells.

## The myristoylated proteome of *Toxoplasma*

To generate a comprehensive list of myristoylated proteins in *Toxoplasma*, we combined YnMyr enrichment with the substrate response to NMT inhibition and the myristoylated peptide discovery. This stringent selection strategy yielded 65 substrates that were further split based on the confidence level (*Supplementary file 4*). 42 substrates were classified as high confidence (the presence of myristoylated peptide and/or robust response to NMT inhibition with YnMyr enrichment ≥ two capture reagents) while 19 were classified as medium confidence (the presence of myristoylated peptide or response to NMT inhibition with YnMyr enrichment with one capture reagent). Finally, PKG and PPM5C that did not pass our confidence criteria, yet have been reported as myristoylated by others (*Brown et al., 2017*; *Yang et al., 2019*), as well as two proteins that responded to NMT inhibition, yet were not present in our global enrichment analysis, were classified as lower confidence hits (*Supplementary file 4*). Our substrate list includes all proteins previously reported as myristoylated, which validates our approach and indicates that this analysis covers a large fraction of the myristoylated proteome in *Toxoplasma*. Notably, in silico prediction for myristoylation (*Alonso et al., 2019*) disagreed with 11 of our high and medium confidence substrates (*Supplementary file 4*). This is not necessarily surprising, given that the prediction was based on a consensus myristoylation sequence derived from other organisms (*Martin et al., 2011*), and highlights the importance of experimental validation of NMT substrates.

90% of our substrate pool represent novel substrates of TgNMT. Several of these proteins have previously been shown to play important functions across the lytic cycle, for example CDPK1 (egress/invasion; [*Lourido et al., 2010*]); PPM5C (attachment; [*Yang et al., 2019*]); ARF1 and Rab5B (trafficking; [*Kremer et al., 2013*; *Liendo et al., 2001*]). Others, for which the precise function has yet to be discovered, were assigned by gene ontology into key functional classes, like kinases, phosphatases, hydrolases and protein binding (*Supplementary file 4*). We did not obtain any evidence for myristoylation on known secreted *Toxoplasma* proteins, such as rhoptry or dense granule proteins, indicating that these are not substrates of host NMT after secretion. Approximately one third of the reported substrates are uncharacterized proteins, indicating that a large amount of myristoylation-related biology is still to be uncovered.

As expected, the identified substrates showed heterogeneous localisation (*Figure 4A*). Utilizing the localisation of organelle proteins by isotope tagging (LOPIT) prediction (*Barylyuk et al., 2020*) within ToxoDB (*Gajria et al., 2008*), we found proteins from key cellular organelles, including the nucleus, mitochondrion, proteasome and micronemes. In agreement with the functional relevance of myristoylation, we found 50% substrates with known or predicted localisation at the plasma membrane (PM), as well as membrane-bound compartments (e.g. inner membrane complex (IMC), endoplasmic reticulum (ER), and Golgi apparatus). Stable attachment at membranes may require a double acylation, that is both myristoylation and palmitoylation (*Wright et al., 2010*), however, only

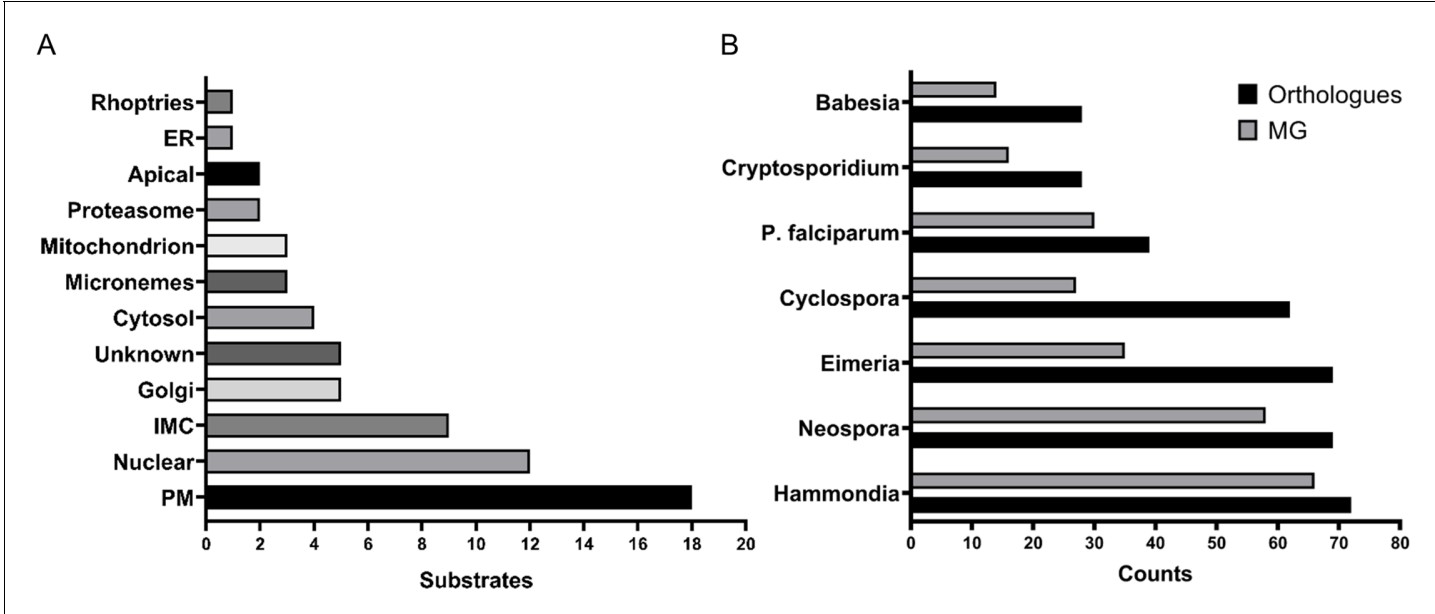

**Figure 4.** The myristoylated proteome of *Toxoplasma gondii*. (**A**) Distribution of the subcellular localisation across our substrate list. Analysis performed using ToxoDB and the build in LOPIT predictor. (**B**) Substrate orthology within selected Apicomplexans. Analysis was performed using EuPathDB. See also *Figure 4—figure supplement 1* and *Supplementary file 4*.

The online version of this article includes the following figure supplement(s) for figure 4:

**Figure supplement 1.** The myristoylated proteome of *Toxoplasma gondii*.

30% of our substrates were previously reported to be palmitoylated (*Caballero et al., 2016*; *Foe et al., 2015*) and *Supplementary file 4*). Since palmitoylation is frequently enriched at the protein N-terminus, in close proximity to the myristate, we analysed the first 20 amino acid sequences of our substrates (*Figure 4—figure supplement 1A*) and found that approximately half possessed cysteine residues (sites of palmitoylation) and, hence, the potential for double acylation. This number correlated well with the 54% palmitoylation prediction (*Ren et al., 2008*) for our substrate pool (*Supplementary file 4*). The reported and predicted palmitoylation data suggested that 12 of the 18 PM substrates likely utilise double acylation for stable membrane attachment while the remaining six may be targeted to the PM via alternative mechanisms. Of the 9 IMC localised substrates, 8 are reported or predicted as palmitoylated as well as 4 of the 5 Golgi-localised ones. This indicates that double acylation is a strong predictor for membrane targeting, albeit to different localisations within the cell, suggesting that further signals are required for their definitive subcellular localisation.

Although the absence of palmitoylation cannot exclude the presence of other secondary signals, such as polybasic regions and PPI sites, which could still aid in PM attachment, we predict that about half of the substrates we identified are likely only myristoylated at the N-terminus. Consistent with this, all cytosolic and proteasome localised substrates were deprived of any palmitoylation and only 3 of the 12 nuclear proteins were shown to be palmitoylated. Within this varied group were CDPK1 (*Ojo et al., 2010*; *Pomel et al., 2008*), the two phosphatases PPM2A and PPM2B (*Yang et al., 2019*) and, surprisingly, the microneme protein MIC7 (*Meissner et al., 2002*). For these proteins myristoylation is likely to serve a distinct function beyond just a simple PM anchor.

The availability of the myristoylated proteomes of *Toxoplasma* and the related *P. falciparum* (*Wright et al., 2014*) allowed us to investigate conserved and non-conserved features of myristoylation across the Apicomplexa. First, we compared both myristoylated proteomes by converting *Plasmodium* myristoylated proteins into *Toxoplasma* orthologues using EuPathDB (*Aurrecoechea et al., 2017*) and compared the overlap of both species. This yielded 24 shared substrates, which corresponds to 37% of the *Toxoplasma* and 63% of the *P. falciparum* experimentally validated myristoylated proteome (*Figure 4—figure supplement 1B*). 39 substrates from the *Toxoplasma* dataset have orthologues in *P. falciparum* and 30 of them contain the MG motif, hinting towards potentially

unexplored *Pf*NMT substrates (*Supplementary file 4*). We also investigated substrate orthology with other Apicomplexans (*Figure 4B*, *Supplementary file 4*). This analysis showed that the lowest level of substrate conservation is present in *Babesia* and *Cryptosporidium* (28 orthologues), followed by *Plasmodium* (39), *Cyclospora* (62), *Eimeria* and *Neospora* (69) and finally *Hammondia* (72). Probing these species-specific orthologues for the presence of the MG motif indicates that between 14 (*Babesia*) and 66 (*Hammondia*) proteins could be potential substrates of NMT (*Figure 4B*, *Supplementary file 4*) and therefore could also be myristoylated in these species. 13 proteins, including CDPK3, PKG, PKAr and ARO, were present in all analysed species, suggesting that their myristoylation may be essential across the phylum (*Supplementary file 4*).

## MIC7 is myristoylated and is important for *Toxoplasma* fitness in vitro

Within our substrate list three proteins were classified as micronemal by LOPIT prediction (*Figure 4A*). TGGT1_249970 was recently described as a protein on the microneme surface where dual acylation is important for its anchoring into the membrane (*Bullen et al., 2016*). The second protein (TGGT1_309990) is annotated as a multi-pass transmembrane protein of unknown function. The third, and perhaps the most interesting, was the microneme protein MIC7 (TGGT1_261780). MIC7 has been reported to be a putative type I transmembrane protein, comprising an N-terminal signal peptide, five EGF-like domains, a membrane-spanning region, and a short cytoplasmic tail (*Meissner et al., 2002*). As MIC signal peptides are typically co-translationally cleaved upon entry into the ER (*Soldati et al., 2001*), the presence of a myristate within the classical signal sequence of MIC7 was unusual. In addition, MIC7 has been suggested to be predominantly expressed in bradyzoites (*Meissner et al., 2002*), the lifecycle stage responsible for the chronic phase of *T. gondii* infection. As our experiments were performed exclusively in tachyzoites, the stage responsible for acute infection, the presence of MIC7 within our dataset could represent a potential false positive identification. To exclude this possibility, we mined MS-based quantification data from an experiment comparing bradyzoite and tachyzoite proteomes (*Young et al., 2020*; PXD019729). The $\log_2$ fold changes in protein abundance for MIC7 and the bradyzoite-specific marker MAG1 (*Tu et al., 2019*; *Figure 5A*, *Supplementary file 5*) revealed that in contrast to MAG1, MIC7 is expressed in tachyzoites, supporting the MS and transcriptional evidence in ToxoDB. We next aimed to directly validate protein myristoylation using ectopically expressed HA-tagged MIC7 WT and myristoylation mutant (Mut, G2G3 > KA) under control of either the endogenous or the strong tubulin promoter. We metabolically labelled parasites with YnMyr and performed a myristoylation-dependent pull down on lysates. Only WT but not the Mut was enriched in this manner (*Figure 5B*), confirming that MIC7 is indeed myristoylated.

To investigate the functional relevance of MIC7 and its myristoylation, we created an inducible knock-out (iKO) line using the DiCre/loxP system (*Andenmatten et al., 2013*) that we recently optimised in RHΔku80 parasites (*Hunt et al., 2019*). The *Mic7* coding sequence was replaced with a floxed, HA-tagged copy of the gene, hereafter called MIC7^HA, that could be excised upon rapamycin (RAPA) treatment (*Figure 5C*). We verified correct integration at the endogenous locus and confirmed RAPA-induced excision by PCR (*Figure 5—figure supplement 1*). At the protein level, MIC7^HA was efficiently depleted 24 hr post RAPA treatment (*Figure 5D*). Correct trafficking of MIC7^HA to micronemes was verified by the co-localisation with the micronemal marker MIC2 (*Figure 5E*). Upon deletion of *Mic7*, parasites no longer formed detectable plaques in host cell monolayers after 5 days in culture, but we could observe very small plaques emerging after 7 days (*Figure 5F*). Collectively these results demonstrate an important, but non-essential, role for MIC7 in the lytic cycle.

## Myristoylation of MIC7 appears important for host cell invasion

To investigate where in the lytic cycle MIC7 plays a role, and test the functional relevance of N-terminal myristoylation, we complemented the iKO line by introducing Ty1-tagged WT or myristoylation defective mutant (hereafter called cWT and cMut, respectively) copies of *Mic7* into the *Uprt* locus (*Figure 6A*). Both inserts were correctly integrated and both complemented lines retained efficient RAPA-induced *Mic7* excision (*Figure 6—figure supplement 1*) and depletion of MIC7^HA (*Figure 6B*). After confirming equivalent and RAPA-insensitive expression of cWT and cMut (*Figure 6B*), we validated both lines in terms of their myristoylation-dependent enrichment and

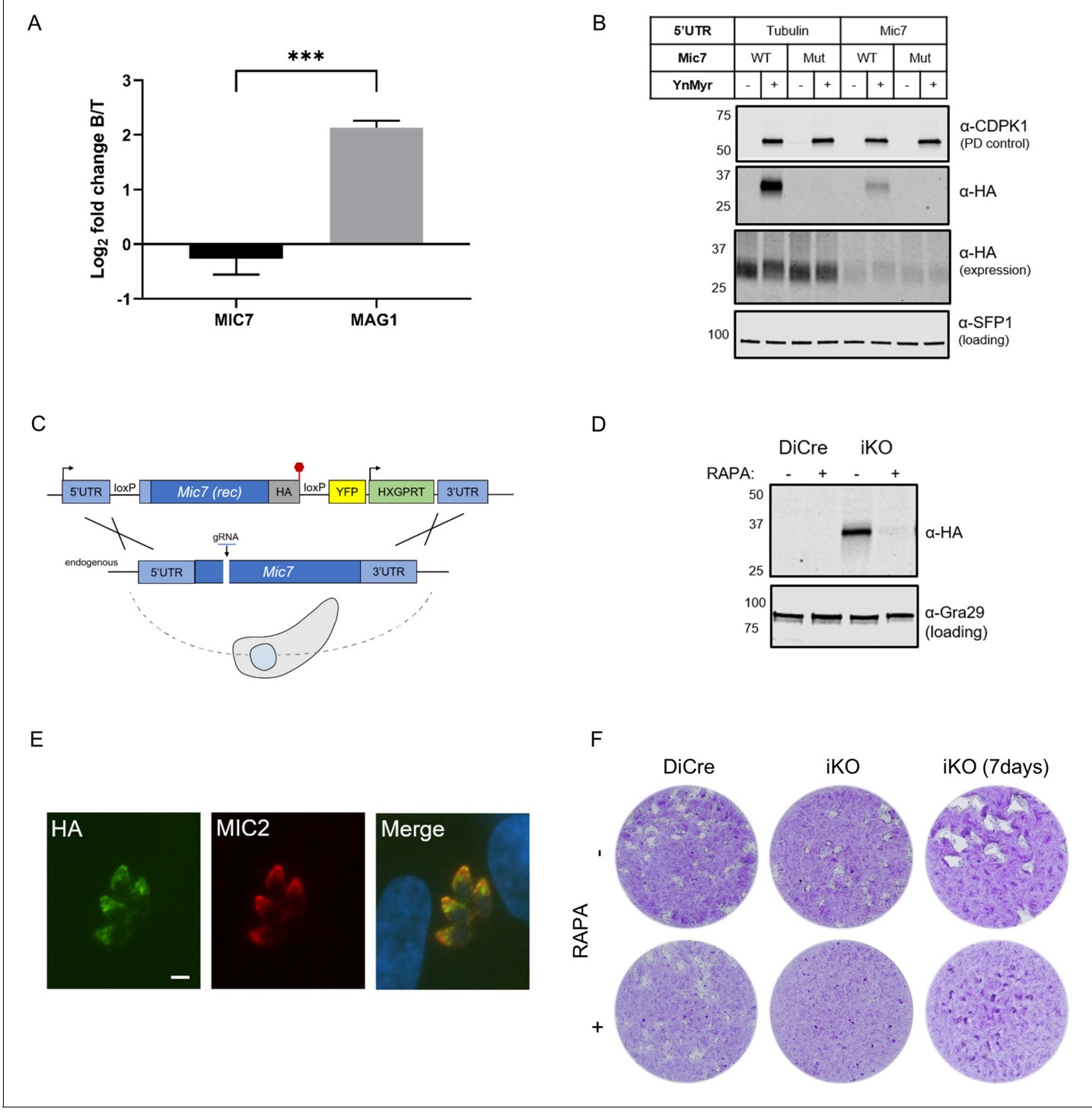

**Figure 5.** MIC7 is myristoylated and is important for *T. gondii* lytic cycle. (**A**) MS-based quantification of MIC7 and MAG1 abundance in tachyzoites [T] and bradyzoites [B] of *T. gondii*. Significance calculated using two-tailed Student's t-test, \*\*\*p=0.0002, N = 3, error bars represent standard deviation. (**B**) MIC7 is myristoylated as shown by YnMyr-dependent pull down and western blotting with α-HA antibody. CDPK1 and SFP1 (TGGT1_289540) are used as enrichment and loading controls, respectively. (**C**) Schematic representation of the DiCre/loxP-based iKO strategy used for the conditional depletion of *Mic7*. Red hexagon represents STOP codon, rec - recodonized. (**D**) Validation of RAPA-dependent depletion of MIC7[HA] in the iKO line illustrated by western blotting with α-HA antibody. Gra29 was used as loading control. (**E**) Co-localisation of MIC7[HA] (green) with the micronemal marker MIC2 (red) in the iKO line by immunofluorescence analysis. Scale bar: 3 μm. (**F**) Plaque assays illustrating that MIC7 is important, but not essential for the intracellular growth of *Toxoplasma*. Assay was performed for 5 days (where not indicated) in three biological replicates, each in technical triplicate, representative images are shown. See also *Figure 5—figure supplement 1*.

*Figure 5 continued on next page*

*Figure 5 continued*

The online version of this article includes the following source data and figure supplement(s) for figure 5:

**Source data 1.** Numerical data of the graph presented in *Figure 5A*.
**Figure supplement 1.** Inducible knock-out of MIC7.

showed that only the cWT was selectively pulled down after metabolic labelling with YnMyr (*Figure 6C*). In the next step, we investigated the co-localisation of MIC7[HA] with cWT and cMut (*Figure 6D*). Both complementation isoforms localised to the micronemes, indicating that the myristate is not required for the trafficking of MIC7 to this organelle.

We next sought to evaluate the role of MIC7 myristoylation in the parasite lytic cycle. While cWT rescued the iKO phenotype upon RAPA treatment, cMut parasites formed substantially smaller plaques under equivalent conditions (*Figure 6E*). This demonstrates that myristoylation indeed plays a key role in MIC7 function. Given the well-established role of microneme proteins in facilitating host cell penetration, we explored whether myristoylation of MIC7 may be important for invasion. We treated iKO, cWT and cMut parasites with RAPA and performed a red/green assay (*Huynh et al., 2003*) which can distinguish invaded from attached parasites. As shown in *Figure 6F*, we observed efficient invasion of host cells by the cWT parasites. This was not the case in the iKO and cMut lines, where invasion was reduced by 57% and 32%, respectively. Compared to the cWT line, we also observed a consistent 61% drop in the total number of iKO parasites (*Figure 6G*), which suggests a defect in the attachment to host cells. A modest but non-significant reduction of 15% in attachment was observed in the cMut strain. Collectively, these results indicate that MIC7 plays an important role in *Toxoplasma* propagation by facilitating parasite attachment and subsequent entry into host cells. Furthermore, myristoylation is not required for sorting MIC7 to the micronemes but appears to be important for its function in invasion of host cells.

## Generation of double tagged MIC7 lines for in depth characterisation of MIC7 myristoylation

To monitor MIC7 N-terminus, and thus the fate of the myristate, we generated double tagged MIC7 variants bearing a Myc tag in the ectodomain and a Ty1 tag at the C-terminus. Placing a Myc tag in the region between the MIC7 transmembrane (TM) domain and the last predicted EGF domain (EGF5) yielded non-functional protein (*Figure 7—figure supplement 1A*). To select a likely suitable position for the tag, we resorted to structural predictions. The region between EGF5 and the TM domain (EGF5-TM, residues 230–284) possesses two pairs of cysteine residues and may represent either an extension of EGF5, or a non-canonical/truncated EGF6. This could explain the inability to place an epitope tag in this region and yield functional MIC7. We also excluded EGF3 and EGF4, that possess the strongest signatures for calcium-binding motifs (PFAM database domain entry PF07645), which normally imparts a rigid domain arrangement with its neighbours. Finally, we considered possible locations (*Figure 7A*) where some degree of structural flexibility would be most likely. The loop between the last two cysteine residues is variable amongst EGF domains, therefore locations were chosen between C38 and C53 of EGF1 and between C86 and C97 of EGF2 as well as within the linker between EGF1 and EGF2. Using ectopic expression, we tested all these positions in terms of protein expression and localisation. All three double tagged protein variants localised to micronemes (*Figure 7—figure supplement 1B*), but the most abundant protein levels were observed when the Myc tag was placed within the EGF2 (*Figure 7—figure supplement 1C*). This indicated that the tag is well tolerated in this position, readily detected by western blot and could be used for further experiments. Myc tagged cWT and cMut (hereafter called [Myc]cWT and [Myc]cMut, respectively) were then inserted into the *Uprt* locus of the iKO line. Both inserts correctly integrated, the new lines retained efficient RAPA-induced *Mic7* excision and depletion of MIC7[HA] (*Figure 7—figure supplement 1D–F*). After verifying equivalent and RAPA independent expression of the [Myc]cWT and [Myc]cMut (*Figure 7—figure supplement 1F*), we confirmed their micronemal localisation by IFA (*Figure 7—figure supplement 1G*). Several microneme proteins have been shown to dimerise (*Cérède et al., 2002*), which harbours the potential that any additional copy of a protein in a merodiploid strain, even when lacking trafficking information, could piggy back as a heterodimer into the micronemes. Indeed, when testing co-IPs in the absence of RAPA, we observed that both

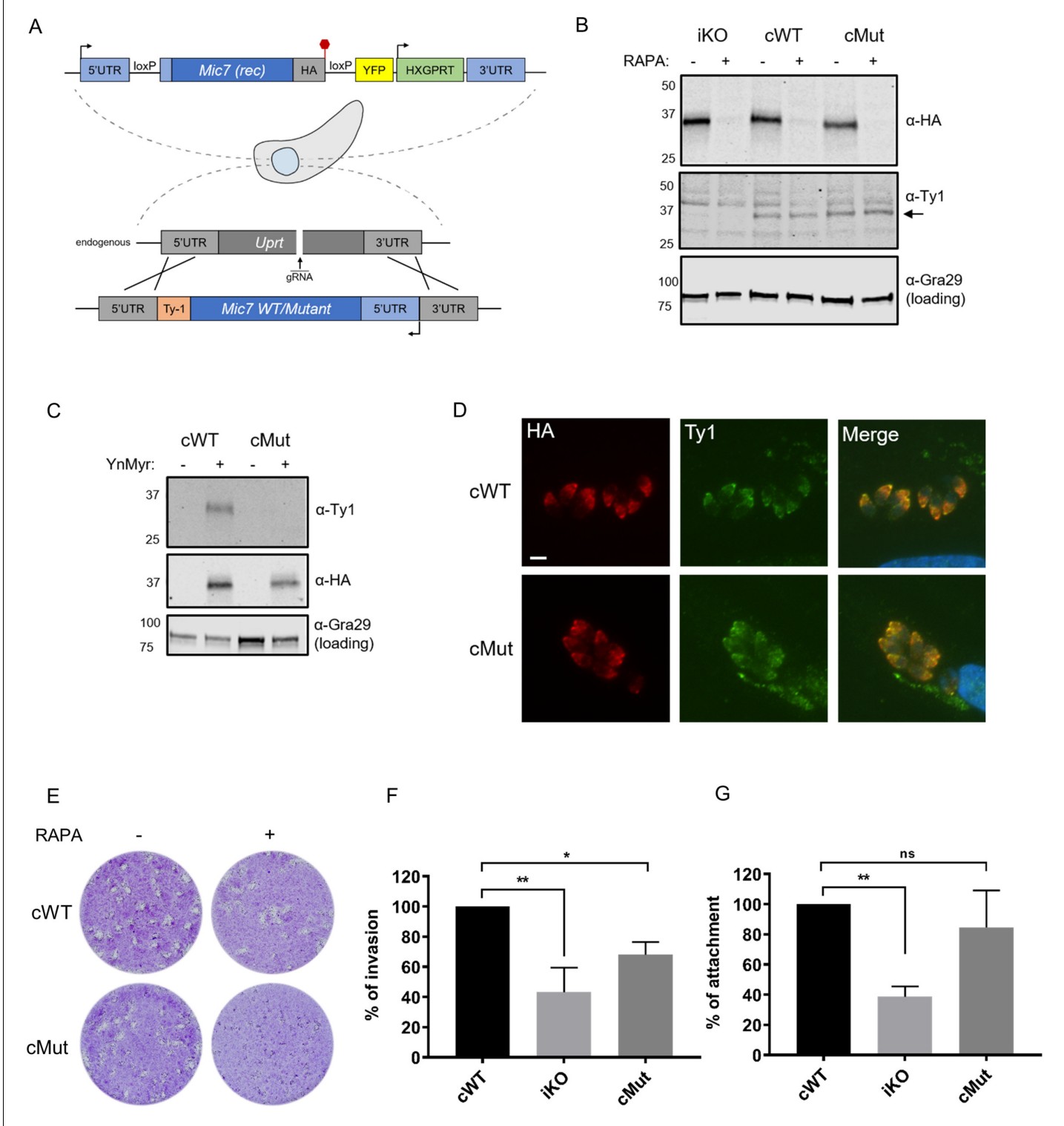

**Figure 6.** Myristoylation of MIC7 plays a role in the invasion of host cells. (**A**) Complementation strategy used to evaluate the functional importance of MIC7 myristoylation. The orientation of cWT and cMut is reversed in the *Uprt* locus with the Ty1 tag at the C-terminus. Red hexagon represents STOP codon, rec – recodonized. (**B**) Western blot analysis demonstrating the RAPA-dependent depletion of MIC7^HA in the iKO, cWT and cMut lines (α-HA) as well as equivalent and RAPA-independent expression of the complements (α-Ty1). Gra29 was used as loading control. (**C**) Biochemical validation of complemented lines by YnMyr-dependent pull down. Enrichment of WT and Mut copy of MIC7 was evaluated by western blotting with α-Ty1 antibody. MIC7^HA (α-HA) and Gra29 were used as enrichment and loading controls, respectively. (**D**) Co-localisation of MIC7^HA (red) with the cWT and cMut (green) by immunofluorescence analysis. Scale bar: 5 μm. (**E**) Plaque assay demonstrating that myristoylation of MIC7 is important in the intracellular

*Figure 6 continued on next page*

*Figure 6 continued*

growth of *Toxoplasma*. Assay performed for 5 days in three biological replicates, each in technical triplicate, representative images are shown. (F) Quantification of invasion efficiency in the RAPA-treated cWT, iKO and cMut lines. Figure shows the average of three biological replicates, each in technical duplicate, error bars represent standard deviation. Significance calculated using 1-way ANOVA with Dunnett's multiple comparison test, **p=0.001, *p=0.018. (G) Quantification of attachment efficiency in the RAPA-treated cWT, iKO and cMut lines. Figure shows the average of three biological replicates, each in technical duplicate, error bars represent standard deviation. Significance calculated using 1-way ANOVA with Dunnett's multiple comparison test, **p=0.004, ns = not significant. See also *Figure 6—figure supplement 1*.

The online version of this article includes the following source data and figure supplement(s) for figure 6:

**Source data 1.** Numerical data of the graphs presented in *Figure 6F and G*.
**Figure supplement 1.** Complementation of the MIC7 iKO line.

complements can co-IP with MIC7^HA (*Figure 7B*), which is independent of the myristate. We therefore repeated the localisation experiments in the presence of RAPA to delete MIC7^HA. As shown in *Figure 7—figure supplement 1H*, both ^Myc^cWT and ^Myc^cMut localise to the micronemes in RAPA-treated parasites, confirming that the myristate is not necessary for MIC7 sorting.

Plaque assays of RAPA-treated complementation lines showed the expected defect in plaque formation for ^Myc^cMut, however, we also observed a small reduction of plaque size for ^Myc^cWT expressing parasites as compared to the DMSO control (*Figure 7C*). This slightly reduced ability to form plaques is likely due to the effect of Myc tag insertion on the MIC7 function. However, since the tag is present in both complemented lines and the ability to form plaques was substantially more impeded in the ^Myc^cMut parasites, these lines are still suitable to investigate the specific function of MIC7 myristoylation.

## MIC7 and its myristoylation are important for the initiation of host cell invasion

In order to shed some light on MIC7 topology within the micronemes as well as the fate of its N-terminal myristate, we performed proteinase K protection assays (*Figure 7D*). In these experiments, proteins/domains that are accessible to proteinase K after digitonin-mediated plasma membrane permeabilisation are digested, while those retained within organelles, such as the micronemes, are protected from this proteolytic digest. We treated tachyzoites from both ^Myc^cWT and ^Myc^cMut lines with RAPA to deplete MIC7^HA and fed with YnMyr to allow for myristoylation-dependent pull down of the complements. We then subjected parasites to digitonin and proteinase K treatment, followed by detection of Myc and Ty1 tags. As a control we used antibodies against the ectodomain of MIC2, as it should be protected (*Bullen et al., 2016*). Under these conditions the MIC7 C-terminus was digested, while the N-terminus was protected as visualised by the Ty1 and the Myc antibodies, respectively. As the parasites were treated with YnMyr prior to the experiment, we could use a YnMyr-dependent pull down to demonstrate that the protected N-terminus of MIC7 remains myristoylated (*Figure 7D*). These results strongly suggest that MIC7 is indeed a transmembrane protein with a myristoylated N-terminus facing the microneme lumen and a short C-terminal cytoplasmic tail that faces the parasite cytoplasm.

Having established the presence of MIC7 N-terminal myristoylation within the micronemes, we aimed to perform a more detailed characterisation of its function. First, we repeated invasion assays in large scale as described in *Touquet et al., 2018*. We performed three independent experiments with a total of 15 replicates per tested parasite line (*Figure 7E*). Taking into account the phenotypic effect for the Myc tag insertion in complemented parasite lines we used untreated iKO parasites as a parental control. Parasites that no longer expressed MIC7^HA displayed a 78% decrease in invasiveness when compared to the control. Complementation with ^Myc^cWT copy can restore the invasiveness to 61%, while ^Myc^cMut reach only 30% invasion capacity. These results are largely consistent with our previous observations (*Figure 6F*) and further confirm the critical role for MIC7 and its myristoylation in the invasion process.

To gain a better understanding of the function of MIC7 during host cell penetration, we filmed invasion of the DMSO and RAPA-treated iKO and the complemented lines into GFP-GPI expressing host cells (*Figure 7—videos 1–4*). We calculated times of successful and failed invasions for each genetic background and observed that despite different success rates between lines, the parasites that did enter the host cell proceed with a similar speed as the control (*Figure 7F*, *Figure 7—videos*

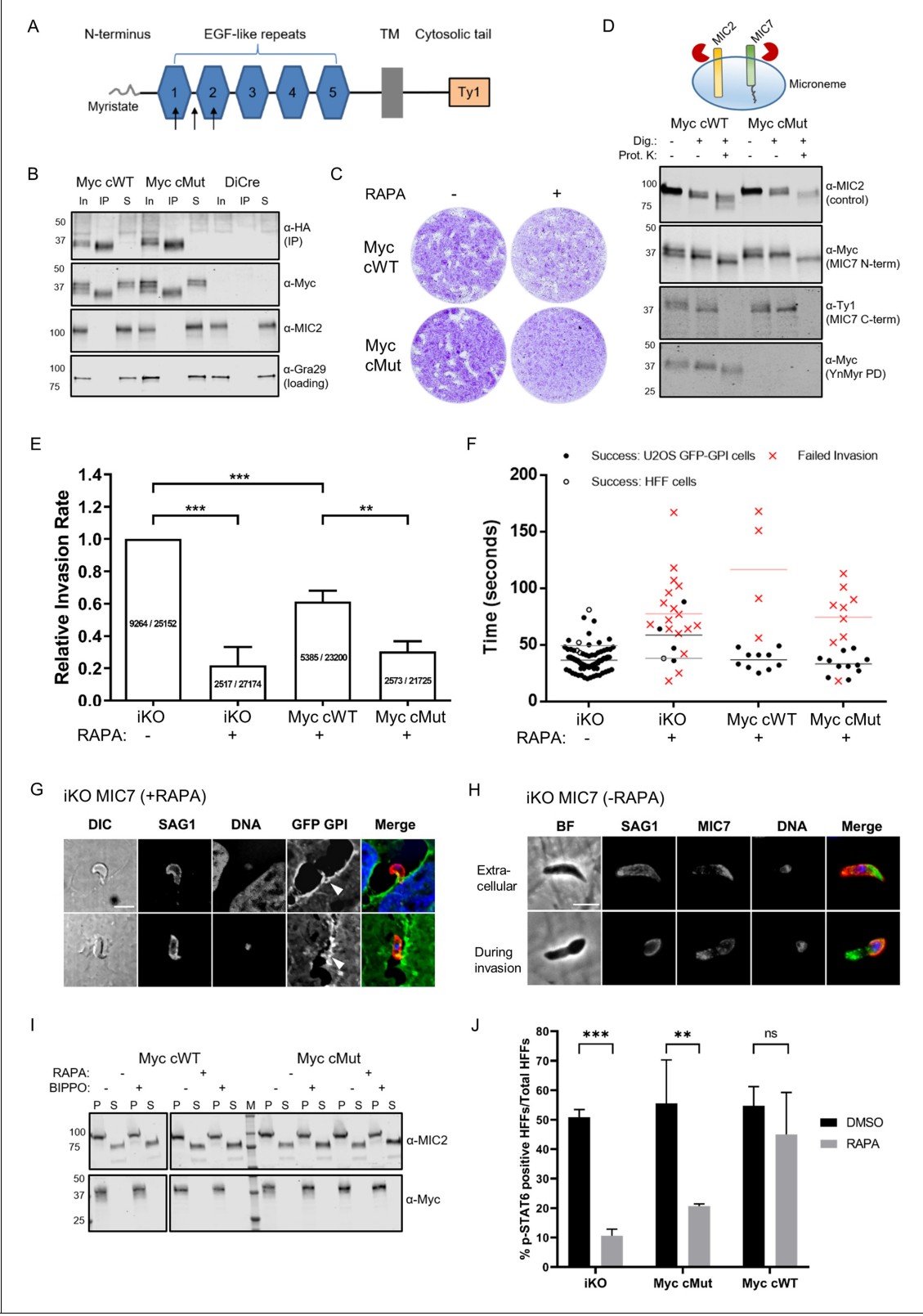

**Figure 7.** Functional analysis of MIC7 and its myristoylation in double tagged MIC7 lines. (**A**) Schematic representation of MIC7 domain structure with evaluated Myc tag positions indicated by arrows. (**B**) Western blot analysis showing co-immunoprecipitation of $^{Myc}$cWT and $^{Myc}$cMut (α-Myc) with MIC7$^{HA}$ (α-HA). MIC2 and Gra29 were used as controls for microneme solubilisation and equal loading, respectively; In = input, IP = immunoprecipitate, S = supernatant after IP. (**C**) Plaque assay confirming that myristoylation of MIC7 is important in the intracellular growth of

*Figure 7 continued on next page*

*Figure 7 continued*

*Toxoplasma* in the newly generated lines. Assay performed for 5 days in three biological replicates, each in technical triplicate, representative images are shown. (D) Proteinase K protection assay demonstrating that the N-terminus of MIC7 is myristoylated and protected (α-Myc) while the C-terminus is digested (α-Ty1) upon permeabilisation and protease treatment. MIC2 was used as a known control where the ectodomain is protected under similar conditions. Dig. = digitonin, Prot.K = proteinase K. (E) Quantitative large scale invasion assay. Results show the relative invasion rate as mean ± standard deviation calculated from three independent assays, each performed with five replicates. The total number of intracellular tachyzoites and human cells are indicated in each column. Two-tailed Student's t-test was performed for two-group comparisons, ***p<0.0006, **p=0.0044. (F) Distribution of the successful/failed invasion times for each of the four genetic backgrounds. Mean invasion time is displayed for both HFF and U2OS host cells. (G) Representative immunofluorescence images (n = 45/68) demonstrating the characteristic bending of the conoid in extracellular RAPA-treated iKO parasites stained against the surface protein SAG1 (marking extracellular parasites and the extracellular part of invading tachyzoites only). It is also possible to observe the U2OS GFP-GPI membrane invagination at the vicinity of the extracellular parasites (white arrowheads). Scale bar: 5 μm. (H) Representative immunofluorescence images of untreated (-RAPA) iKO parasites while extracellular (n = 63) and during invasion (n = 22). Parasites were stained against SAG1 (marking extracellular parasites and the extracellular part of invading tachyzoites only) and α-HA to stain MIC7. Scale bar: 5 μm. (I) Analysis of MIC7 secretion upon stimulation with BIPPO. No shedding of MIC7 was observed (α-Myc) in contrast to MIC2 which was used as a control. P - pellet, S - supernatant containing the excreted secreted antigens, M – molecular weight marker. White space was used to indicate where gel lanes were not contiguous. (J) Immunofluorescence based quantification of pSTAT6 as a reporter for ROP16 secretion in all tested lines with and w/ o RAPA treatment. Figure shows the average of three biological replicates, error bars represent standard deviation. Significance calculated using 2-way ANOVA with Sidak's multiple comparison test, ***p=0.0004, **p=0.0013, ns = not significant. See also *Figure 7—figure supplements 1* and *2*.

The online version of this article includes the following video, source data, and figure supplement(s) for figure 7:

**Source data 1.** Numerical data of the graph presented in *Figure 7E*.
**Source data 2.** Numerical data of the graph presented in *Figure 7F*.
**Source data 3.** Numerical data of the graph presented in *Figure 7J*.
**Figure supplement 1.** Generation of doubly-tagged MIC7 lines.
**Figure supplement 2.** Functional analysis of MIC7 and its myristoylation.
**Figure 7—video 1.** Related to *Figure 7F*.
https://elifesciences.org/articles/57861#fig7video1
**Figure 7—video 2.** Related to *Figure 7F*.
https://elifesciences.org/articles/57861#fig7video2
**Figure 7—video 3.** Related to *Figure 7F*.
https://elifesciences.org/articles/57861#fig7video3
**Figure 7—video 4.** Related to *Figure 7F*.
https://elifesciences.org/articles/57861#fig7video4

*1–4*). This suggests a potential failure to initiate invasion. In several failed invasion events we also observed a notable membrane invagination (*Figure 7—figure supplement 2A*, *Figure 7—videos 2* and *4*), which normally occurs after secretion of the rhoptry neck components required for the formation of the tachyzoite-host cell junction (*Bichet et al., 2016*). It is worth noting, that the immunofluorescent analysis of likely aborted invasion events in the RAPA treated iKO line revealed more than half of extracellular parasites (45 of 68) show a strong arch-shaped morphology while associated with a host cell (*Figure 7G*). This is indicative of MIC7 KO parasites exerting force during the attempt to invade, possibly leading to the zoite deformation.

The inability to initiate invasion in the iKO and ^MyccMut lines could be caused by a general defect in gliding motility or a failure to secrete micronemes. However, both circular and helical trails were detected with no obvious differences between parasites (*Figure 7—videos 1–4*, *Figure 7—figure supplement 2B*). We also observed no reduction in microneme secretion, as shown by efficient MIC2 processing in MIC7 KO parasites, even in the presence of 5-benzyl-3-isopropyl-1H-pyrazolo [4,3-d]pyrimidin-7(6H)-one (BIPPO), a phosphodiesterase inhibitor that triggers signalling pathways leading to increased parasite motility and microneme secretion (*Figure 7—figure supplement 2C*).

To get a better understanding of MIC7 distribution in parasites before and during invasion, we co-localised MIC7 with SAG1, a well characterized surface marker, which can be used to delineate the intra- and extracellular part of invading parasites. This revealed that MIC7 partially redistributes around the periphery of extracellular and invading parasites, but appears absent from the tachyzoite-host cell junction (*Figure 7H*), a constriction through which the parasite moves as it enters the host cell (*Pavlou et al., 2018*; *Tyler et al., 2011*). This redistribution suggests, but is not proof, that MIC7, as many other microneme proteins, is secreted onto the parasite surface during or after egress from the host cell. Most transmembrane microneme proteins undergo proteolytic maturation

near or within the transmembrane domain after egress and during invasion (*Soldati et al., 2001*). This process is facilitated by subtilisin or rhomboid proteases and is thought to relieve the high affinity interactions between the parasite and host cell receptors (*Carruthers, 2006*; *Dowse and Soldati, 2005*). We analysed if MIC7 could undergo similar processing and found that while MIC2 was efficiently cleaved and released into the culture supernatant, this was not the case for MIC7, even in the presence of BIPPO (*Figure 7I*). We also tested whether MIC7 could be shed during invasion by analysing the culture supernatant as well as parasite and host material of freshly invaded cells. Again, no MIC7 shedding was observed (*Figure 7—figure supplement 2D*). These results are in agreement with previously reported observations (*Meissner et al., 2002*), and confirm that MIC7 is not subject to a proteolytic cleavage.

## MIC7 and its myristoylation may be important for efficient secretion of rhoptry contents into the host cell

The live imaging data of the iKO parasites suggested that MIC7 plays an important role in the onset of invasion, potentially after the establishment of the tachyzoite-host cell junction. In one instance we could observe a membrane swirl formation at the apex of the parasite attempting invasion, suggesting that rhoptry contents may have been secreted (*Figure 7—video 2*). Rhoptries are apical organelles which contain a number of kinases the parasite injects into the host cell upon invasion. These effector kinases modulate many important functions in host-microbe interaction (*Boothroyd and Dubremetz, 2008*). One of these kinases, the rhoptry kinase 16 (ROP16) acts as a JAK mimetic leading to rapid and direct phosphorylation of STAT6 (*Ong et al., 2010*; *Saeij et al., 2007*). Accordingly, we used STAT6 phosphorylation as a reporter for efficient rhoptry secretion and ROP16 translocation into the host cell (*Figure 7J*). RAPA-treated $^{Myc}$cWT parasites showed 45% pSTAT6 positive nuclei. RAPA-treated iKO and $^{Myc}$cMut parasites showed a significant reduction of pSTAT6 positive nuclei (11% and 21%, respectively). No significant differences in pSTAT6 positive cells between tested parasite lines were observed when treated with DMSO. This indicates that iKO and $^{Myc}$cMut parasites can still secrete ROP16 into the host cell and induce STAT6 phosphorylation, although at significantly reduced levels.

## Discussion

Our understanding of myristoylation and its functional consequences in *Toxoplasma* is hampered by the limited knowledge of NMT substrates. Using an integrated MS-approach we describe here the first experimentally validated myristoylated proteome in *T. gondii*. We combine two orthogonal chemoproteomic techniques, that is quantitative response to NMT inhibition with direct MS/MS evidence for substrate modification, which allows for high confidence in substrate identification as well as substantial substrate coverage. Despite the complex nature of our samples, consisting of both human and parasite proteins, our discovery includes all proteins previously reported to be myristoylated in *Toxoplasma* as well as novel and unexpected *Tg*NMT substrates. The fact that these proteins are functionally diverse, and involved in all steps of the lytic cycle highlights the importance of myristoylation in *Toxoplasma* biology. Consistent with this, treatment with NMTi resulted in dose dependent parasite killing. Although we cannot exclude small off-target effects of IMP-1002 that was originally designed for *Plasmodium spp.*, we predict that this severe phenotype was largely due to pleiotropic effects of *Tg*NMT inhibition on the *Toxoplasma* lytic cycle. Potent and selective *Tg*NMT inhibitors are yet to be reported, however extensive work in other protozoan parasites (*Ritzefeld et al., 2018*) demonstrates that selective NMT inhibition could provide an attractive strategy to combat infection. Moreover, results presented here indicate that related parasites, with high structural conservation of their NMTs, could be inhibited by a single compound, which may allow for the development of a pan-parasite inhibitor in the future.

Our substrates showed heterogeneous localisation with ca. 50% localised to PM or membrane bound compartments. Palmitoylation analysis confirmed that for the majority of these substrates stable attachment to membranes is likely driven by double acylation. Although we cannot exclude the presence of other secondary signals which could aid in PM targeting within our substrate pool, such as polybasic regions and PPI sites, our analysis and the predicted localisation suggest that many of our substrates may be myristoylated only, indicating that their myristoylation can serve more discrete functions than just a priming site for the palmitate. Such alternate functions could include reversible

membrane binding by the conformation regulated exposure of the myristate as shown for mammalian ARF1 (*Goldberg, 1998*), regulation of protein activity as demonstrated for GPAT4 during glycerolipid synthesis (*Zhu et al., 2019*), or involvement in PPIs, as shown for the viral capsid assembly (*Chow et al., 1987*; *Mousnier et al., 2018*).

Here we identified unexpected myristoylation of the *Toxoplasma* microneme protein 7 (MIC7). Microneme proteins are key factors in *Toxoplasma* propagation involved in parasite egress, motility and host cell invasion. They are trafficked into the secretory pathway by virtue of an N-terminal signal peptide that is cleaved during ER import (*Soldati et al., 2001*). We show that MIC7 is an important, yet not essential protein and we unequivocally demonstrate that MIC7 is myristoylated at its N-terminus, excluding the existence of an N-terminal signal sequence. While it is known that proteins can enter the secretory pathway by virtue of a recessed signal or leader peptide, this has not been reported for microneme proteins. Furthermore, many known microneme proteins are type I transmembrane proteins, where an N-terminal PPI or carbohydrate binding domain faces the microneme lumen and a short cytoplasmic domain faces the parasite cytosol. Upon microneme secretion the protein is transferred to the parasite PM, with the interacting domain exposed to bind to host cell receptors. Because MIC7 lacks a signal peptide but otherwise mimics this domain structure, it most likely enters the secretory pathway as a type III transmembrane protein (*Goder and Spiess, 2001*). Although we have not specifically tested this here, this is supported by several features in the MIC7 primary amino acid sequence. That is: a transmembrane domain of > 20 amino acids in length (22 in MIC7) and positive charges on the C-terminal side of the transmembrane domain.

We demonstrated that MIC7 and its myristoylation are important in host cell invasion. Live imaging of the invasion process further revealed a potential defect in initiating invasion in the MIC7 KO and myristoylation defective parasites. This phenomenon was not caused by a fault in microneme secretion itself or parasite gliding. The live video analysis suggests that the parasites may be able to initiate the tachyzoite-host cell junction (supported by some rhoptry content secretion as measured by STAT6 phosphorylation), but then fail to progress beyond the initiation. This phenotype is similar to *Toxoplasma* Myosin A knockout parasites, that are also able to initiate invasion, but then fail to invade given the lack of a functional actin-myosin system. However, the strong arch-shaped deformation of extracellular MIC7 KO parasites in live invasion assays is substantially different from the MyoA knockouts (*Bichet et al., 2016*). While we could not unequivocally show that MIC7 is secreted to the zoite surface to engage with host cell receptors via its EGF domains, the fact that it redistributes around the periphery of the cell shortly after egress and during invasion supports this scenario. When secreted, the myristate could contribute to specific PPIs with other surface proteins or potentially host ligands to support attachment and initiate invasion.

It is also tempting to speculate that alternatively, the myristate could be inserted into the lipid bilayer of the host cell. It has been shown that HBV viruses can utilise a myristoylated protein on their surface to enter host cells (*Maurer-Stroh and Eisenhaber, 2004*). In *Toxoplasma* this mechanism could contribute to host cell attachment, for which we observed a measurable defect upon MIC7 deletion, but also to the earliest events of invasion, that is rhoptry secretion or host-cell penetration. This is reminiscent of MIC8, another microneme protein, that has been implicated in a signalling cascade leading to rhoptry discharge (*Kessler et al., 2008*). Our experiments to probe secretion of ROP16, which rapidly phosphorylates STAT6 (*Ong et al., 2010*; *Saeij et al., 2007*), indicate that rhoptry secretion per se is not affected upon MIC7 deletion, but reduced ROP16 mediated STAT6 phosphorylation is observed. We cannot currently distinguish whether the decrease in ROP16 injected cells is due to decreased invasion efficiency of MIC7 KO and $^{Myc}$cMut lines, or whether equal amounts of cells have been attempted to be invaded but ROP16 has not been efficiently transferred. However, the attachment phenotype observed for MIC7 myristoylation mutants is less profound than the reduction of pSTAT6 induction, suggesting that the major cause of the rhoptry secretion phenotype is independent of attachment and most likely due to an inefficient translocation of rhoptry contents into the host cell. Whether the failure to progress beyond the initiation of the tachyzoite-host cell junction lies in a failure to properly secrete rhoptry contents, or to form the junction itself remains to be clarified.

In summary we provide a useful resource of experimentally validated myristoylated and GPI-anchored proteins, as well as first clues to the identity of so far uncharacterized myristoylated proteomes across the phylum. We show that an NMT inhibitor that was generated against *Plasmodium spp.* also inhibits *Toxoplasma* growth. The presence of several essential *N*-myristoylated proteins

conserved across many Apicomplexa indicates that NMT inhibition by a single compound may be a viable strategy to target several pathogens. We have identified N-terminal myristoylation on a *Toxoplasma* protein that uses an unconventional mode of trafficking to the parasite secretory pathway and the micronemes, and displays a novel use of myristoylation in parasite biology. This unexpected discovery, which can likely be found in other organisms, demonstrates how our dataset can serve as a tool in target-specific investigations that can ultimately help to unravel the exciting biology of host-microbe or more broadly, cell-cell interactions.

# Materials and methods

## Key resources table

| Reagent type (species) or resource | Designation | Source or reference | Identifiers | Additional information |
|---|---|---|---|---|
| Synthetic Gene (*Toxoplasma gondii*) | Microneme protein 7 (MIC7) | GeneArt, Life Technologies | TGGT1_261780 (http://toxodb.org) | Floxed and HA tagged sequence |
| Cell line (*Homo sapiens*) | Human foreskin fibroblasts (HFFs) | ATCC | Cat# SCRC-1041, RRID:CVCL_3285 | The cell line is available from the American Type Culture Collection (ATCC) |
| Cell line (*Toxoplasma gondii*) | RH Δ*ku80* Δ*hxgprt* | Huynh and Carruthers, 2009 | | Used in all mass spectrometry experiments |
| Cell line (*Toxoplasma gondii*) | RH DiCre Δ*ku80* Δ*hxgprt* | Hunt et al., 2019 | | The second-generation DiCre-expressing cell line in *Toxoplasma gondii* |
| Cell line (*Toxoplasma gondii*) | iKO MIC7; RH DiCre Δ*ku80* Δ*hxgprt_LoxMic7_HA* | This paper | | The endogenous *Mic7* gene was replaced with a floxed and HA-tagged *Mic7* gene |
| Cell line (*Toxoplasma gondii*) | cWT MIC7 | This paper | | As described for the iKO MIC7 line, however a *Mic7-Ty1* expressing construct was integrated into the *UPRT* locus. |
| Cell line (*Toxoplasma gondii*) | cMut MIC7 | This paper | | As described for the iKO MIC7 line, however a *Mic7(G2K/G3A)-Ty1* expressing construct was integrated into the *UPRT* locus. |
| Cell line (*Toxoplasma gondii*) | Myc cWT MIC7 | This paper | | As described for the cWT MIC7 line, however a *Myc-Mic7-Ty1* expressing construct was integrated into the *UPRT* locus. |
| Cell line (*Toxoplasma gondii*) | Myc cMut MIC7 | This paper | | As described for the cMut MIC7 line, however a *Myc-Mic7(G2K/G3A)-Ty1* expressing construct was integrated into the *UPRT* locus. |
| Antibody | Rat anti-HA monoclonal clone 3F10 | Roche | Cat# 11867423001 RRID:AB_390919 | WB (1:1000) IFA (1:1000) |
| Antibody | Mouse anti-Myc monoclonal clone 4A6 | Millipore | Cat# 05–724 RRID:AB_11211891 | WB (1:1000) IFA (1:1000) |

*Continued on next page*

*Continued*

| Reagent type (species) or resource | Designation | Source or reference | Identifiers | Additional information |
|---|---|---|---|---|
| Antibody | Mouse anti-Ty1 monoclonal clone BB2 | Thermo Fisher | Cat# MA5-23513 RRID:AB_2610644 | WB (1:2000) IFA (1:500) |
| Antibody | Mouse anti-*Toxoplasma* monoclonal clone TP3 | Abcam | Cat# ab8313 RRID:AB_306466 | WB (1:1000) |
| Antibody | Mouse anti-MIC2 monoclonal clone 6D10 | other | | Provided by Vernon Carruthers Lab WB (1:1000) |
| Antibody | Rabbit anti-MIC2 polyclonal | other | | Provided by Vernon Carruthers Lab WB (1:500) IFA (1:5000) |
| Antibody | Rabbit anti-*Tg*CAP polyclonal | *Hunt et al., 2019* | | WB (1:2000) |
| Antibody | Rabbit anti-Gra29 polyclonal | *Young et al., 2020* | | WB (1:1000) |
| Antibody | Rabbit anti-SFP1 polyclonal | *Young et al., 2020* | | WB (1:1000) |
| Antibody | Mouse anti-CDPK1 polyclonal | other | | Provided by Matthew Child and Matt Bogyo WB (1:3000) |
| Antibody | Rabbit anti-SAG1 monoclonal | other | | Provided by John Boothroyd Lab WB (1:10,000) |
| Antibody | Rabbit anti-GAP45 polyclonal | other | | Provided by Peter Bradley Lab WB (1:1000) |
| Antibody | Rabbit anti-phospho-Stat6 polyclonal | Cell Signaling | Cat# 9361 RRID:AB_331595 | IFA (1:600) |
| Chemical compound | Myristic acid (Myr) | Tokyo Chemical Industry | Cat# M0476 | |
| Chemical compound | Alkyne-myristic acid (YnMyr) | Iris Biotech | Cat# RL-2055 | |
| Chemical compound | Azide-PEG3-biotin (N3-biotin) | Sigma-Aldrich | Cat# 762024 | Capture reagent 1 |
| Chemical compound | Trypsin cleavable reagent | *Broncel et al., 2015* | | The reagent used here was synthesised in-house by the Peptide Chemistry science technology platform, The Francis Crick Institute |
| Chemical compound | TEV cleavable reagent | *Speers and Cravatt, 2005* | | The reagent used here was synthesised in-house by the Peptide Chemistry science technology platform, The Francis Crick Institute |
| Chemical compound | IMP-1002 | *Schlott et al., 2019* | | The reagent used here was synthesised by the Tate Laboratory, Imperial College London |
| Chemical compound | Rapamycin | Sigma-Aldrich | Cat# R8781 | |

*Continued on next page*

*Continued*

| Reagent type (species) or resource | Designation | Source or reference | Identifiers | Additional information |
|---|---|---|---|---|
| Chemical compound | 5-Benzyl-3-isopropyl-1H-pyrazolo[4,3-d]pyrimidin-7(6H)-one (BIPPO) | *Howard et al., 2015* | | The reagent used here was synthesised in-house by the Peptide Chemistry science technology platform, The Francis Crick Institute |
| Software, algorithm | MaxQuant (version 1.5.0.25 and 1.5.2.8) | *Cox and Mann, 2008* | RRID:SCR_014485 | Free software for searching of mass spectrometry acquisition files |
| Software, algorithm | Perseus (version 1.5.0.9) | *Tyanova et al., 2016* | RRID:SCR_015753 | Free software for processing of MaxQuant output files |
| Software, algorithm | PyMOL (version 1.3r1) | Schrodinger LLC | RRID:SCR_000305 | Commercial software for molecular visualisation |
| Software, algorithm | Prism 8 (version 8.1.1) | GraphPad Software, Inc | RRID:SCR_002798 | Commercial software for statistical analysis |

## General

Reagents: $CuSO_4$, TCEP, TBTA, buffer salts, DTT, iodoacetamide, DMSO, BSA, Triton-X100 and Tween-20 were from Sigma-Aldrich. Azide-PEG3-biotin was from Sigma-Aldrich. Peptide synthesis coupling reagents HATU and HCTU were from Fluorochem and Merck, respectively. MS-grade water, acetonitrile, methanol, TFA and formic acid were from Thermo Scientific. IMP-1002 was synthesised as described in *Schlott et al., 2019*. BIPPO was synthesised as described in *Howard et al., 2015*.

## Plasmid generation

Primers used throughout this study are listed in *Supplementary file 6*. Plasmid sequences were confirmed by Sanger sequencing (Eurofins Genomics).

To generate the *Mic7* iKO plasmid, pG140_MIC7_HA_iKO_loxP100, the *Mic7* 5'UTR with a *loxP* site inserted 100 bp upstream of the *Mic7* start codon, and a recodonised *Mic7* cDNA-HA sequence, were synthesized (GeneArt strings, Life Technologies). These DNA fragments were Gibson cloned into the ApaI/PacI digested parental vector p5RT70loxPKillerRedloxPYFP-HX (*Andenmatten et al., 2013*) to generate an intermediate plasmid. The *Mic7* 3'UTR was subsequently amplified from genomic DNA using primers 1 and 2, while *mCherry* flanked by *Gra* gene UTRs was amplified from pTKO2C (*Caffaro et al., 2013*) using primer pair 3/4. The resulting fragments were Gibson cloned into the SacI-digested intermediate plasmid to generate pG140_MIC7_HA_iKO_loxP100.

To generate the complementation construct pUPRT_MIC7_Ty1, the *Mic7* sequence flanked by its 5'UTR was amplified from genomic DNA using primer pair 5/6. In parallel, the *Uprt* targeting vector pUPRT_HA (*Reese et al., 2011*) was amplified by inverse PCR using primers 7 and 8. The resulting PCR amplicons were Gibson cloned to generate pUPRT_MIC7_Ty1. Primers 5 and 8 comprise overhangs to facilitate introduction of a Ty1 tag 3' of the *Mic7* sequence.

To generate the complementation construct pUPRT_MIC7(G2K/G3A)_Ty1, the *Mic7* 5'UTR and *Mic7* endogenous sequence were amplified using primer pairs 9/10 and 5/11, respectively. In parallel, the *Uprt* targeting vector pUPRT_HA (*Reese et al., 2011*) was amplified by inverse PCR using primers 7 and 8. The resulting PCR amplicons were Gibson cloned to generate pUPRT_MIC7(G2K/G3A)_Ty1. Primers 9 and 11 comprise overhangs that introduce point mutations G2K and G3A, while primers 5 and 8 introduce a Ty1 tag 3' of the *Mic7* sequence.

To generate the complementation construct pUPRT_Myc_MIC7_Ty1, the Myc tag coding sequence was introduced within pUPRT_MIC7_Ty1 plasmid by inverse PCR using primers 12 and 13. The resulting linear fragment was circularized using KLD reaction buffer (NEB) as per manufacturer's instructions.

To generate the complementation construct pUPRT_Myc_MIC7(G2K/G3A)_Ty1, the Myc tag coding sequence was introduced within pUPRT_MIC7(G2K/G3A)_Ty1 plasmid by inverse PCR using primers 12 and 13. The resulting linear fragment was circularized using KLD reaction buffer (NEB) as per manufacturer's instructions.

To generate pSag1_Cas9-U6_sgMIC7, the pSag1_Cas9-U6_sgUPRT (*Shen et al., 2014*; Addgene plasmid # 54467) vector was amplified by inverse PCR using primers 14 and 15. Primer 15 comprises a sequence extension that replaces the *Uprt*-targeting sgRNA with a sgRNA sequence targeting *Mic7*. The resulting linear fragment was circularized using KLD reaction buffer (NEB) as per manufacturer's instructions.

To generate pGra_5'UTRMIC7_MIC7_HA, the 5'UTR of *Mic7* was amplified from gDNA using primer pair 30/31, and recodonised *Mic7* sequence was amplified from pG140_MIC7_HA_iKO_-loxP100 using primers 32 and 33. In parallel, the vector pGra_ApiAT5-3_HA (*Wallbank et al., 2019*) was amplified by inverse PCR using primer pair 34/35. The three resulting PCR amplicons were Gibson assembled to generate pGra_5'UTRMIC7_MIC7_HA.

To generate pGra_5'UTRMIC7_MIC7(G2K/G3A)_HA, the 5'UTR of *Mic7* was amplified from gDNA using primer pair 30/31, and recodonised *Mic7(G2K/G3A)* sequence was amplified from pG140_MIC7_HA_iKO_loxP100 using primers 36 and 33. Primer 36 was used to introduce the point mutations G2K and G3A into the *Mic7* recodonised sequence. In parallel, the vector pGra_ApiAT5-3_HA (*Wallbank et al., 2019*) was amplified by inverse PCR using primer pair 34/35. The three resulting PCR amplicons were Gibson assembled to generate pGra_5'UTRMIC7_MIC7(G2K/G3A)_HA.

To generate pGra_5'UTRTUB_MIC7_HA, the *Tub* 5'UTR was amplified from gDNA using primer pair 37/38, and recodonised *Mic7* sequence was amplified from pG140_MIC7_HA_iKO_loxP100 using primers 39 and 33. In parallel, the vector pGra_ApiAT5-3_HA (*Wallbank et al., 2019*) sequence was amplified by inverse PCR using primer pair 34/40. The three resulting PCR amplicons were Gibson assembled to generate pGra_5'UTRTUB_MIC7 _HA.

To generate pGra_5'UTRTUB_MIC7(G2K/G3A)_HA, the *Tub* 5'UTR was amplified from gDNA using primer pair 37/38, and recodonised *Mic7(G2K/G3A)* sequence was amplified from pG140_MIC7_HA_iKO_loxP100 using primers 41 and 33. Primer 41 was used to introduce the point mutations G2K and G3A into the *Mic7* recodonised sequence. In parallel, the vector pGra_ApiAT5-3_HA (*Wallbank et al., 2019*) was amplified by inverse PCR using primer pair 34/40. The three resulting PCR amplicons were Gibson assembled to generate pGra_5'UTRTUB_MIC7(G2K/G3A) _HA.

To generate pGra_5'UTRMIC7_Myc1_MIC7_HA, pGra_5'UTRMIC7_Myc1/2_MIC7_HA and pGra_5'UTRMIC7_Myc2_MIC7_HA, the Myc tag coding sequence was introduced at different positions within pGra_5'UTRMIC7_MIC7_HA plasmid by inverse PCR using primers 42 and 43 (Myc1), 44 and 45 (Myc1/2), and 46 and 47 (Myc2). The resulting linear fragments were circularized using KLD reaction buffer (NEB) as per manufacturer's instructions.

## Parasite line generation

Freshly harvested parasites were transfected by electroporation (1500 V) using the Gene Pulser Xcell system (Bio-Rad) as previously described (*Soldati and Boothroyd, 1993*).

To generate the inducible MIC7 knock-out line (RH DiCreΔku80Δhxgprt_loxPMIC7_HA, referred to here as iKO MIC7), the plasmid pG140_MIC7_HA_iKO_loxP100 was linearized using PciI and co-transfected with pSag1_Cas9-U6_sgMIC7 into the RH DiCreΔku80Δhxgprt line (*Hunt et al., 2019*). Recombinant parasites were selected 24 hr post transfection by addition of mycophenolic acid (MPA; 25 µg/mL) and xanthine (XAN; 50 µg/mL) to culture medium. Resistant non-fluorescent parasites were cloned, and successful 5' and 3' integration at the *Mic7* locus was confirmed using primer pairs 16/17 and 18/19, respectively. Absence of the endogenous *Mic7* locus was confirmed using primers 24 and 25. Rapamycin-induced excision of the *loxPMic7* sequence was confirmed using primer pair 26/27.

To complement the iKO MIC7 line with MIC7-expressing constructs, pUPRT_MIC7_Ty1 and pUPRT_MIC7(G2K/G3A)_Ty1 or pUPRT_Myc_MIC7_Ty1 and pUPRT_Myc_MIC7(G2K/G3A)_Ty1 plasmids were linearized with ScaI and individually co-transfected with pSAG1_Cas9-U6_sgUPRT. Transgenic parasites were subjected to 5'-fluo-2'-deoxyuridine (FUDR) selection (5 µM) 24 hr post transfection. Resistant parasites were cloned, and successful 5' and 3' integration was confirmed

using primer pairs 20/21 and 22/23. Disruption of the endogenous *Uprt* locus was confirmed using primer pair 28/29.

To generate lines that express WT and myristoylation mutant (G2K/G3A) MIC7 ectopically, plasmids pGra_5'UTRMIC7_MIC7_HA, pGra_5'UTRMIC7_MIC7(G2K/G3A)_HA, pGra_5'UTRTUB_MIC7_HA, and pGra_5'UTRTUB_MIC7(G2K/G3A)_HA were linearized using NotI and individually transfected into the RH Δhxgprt strain. Recombinant parasites were selected 24 hr post transfection by addition of mycophenolic acid (MPA; 25 µg/mL) and xanthine (XAN; 50 µg/mL) to culture medium.

To generate lines that ectopically express MIC7 with Myc tag within the EGF1, EGF1/2 or EGF2 domains, plasmids pGra_5'UTRMIC7_Myc1_MIC7_HA, pGra_5'UTRMIC7_Myc1/2_MIC7_HA and pGra_5'UTRMIC7_Myc2_MIC7_HA were linearized using NotI and individually transfected into the RH Δhxgprt strain. Recombinant parasites were selected 24 hr post transfection by addition of mycophenolic acid (MPA; 25 µg/mL) and xanthine (XAN; 50 µg/mL) to culture medium.

## Cell culture

Parasites of the RH strain were cultured in Human foreskin fibroblasts (HFFs) monolayers in Dulbecco's Modified Eagle Medium (DMEM), GlutaMAX (Thermo Fisher) supplemented with 10% heat-inactivated foetal bovine serum (FBS; Life technologies), at 37°C and 5% $CO_2$. All strains and host cell lines tested negative for the presence of mycoplasma.

## Metabolic labelling and cell lysis

Upon infection of HFF monolayers the medium was removed and replaced by fresh culture medium supplemented with 25 µM YnMyr (Iris Biotech) or Myr (Tokyo Chemical Industry). The parasites were then incubated for 16 hr, washed with PBS (2x) and lysed on ice using a lysis buffer (PBS, 0.1% SDS, 1% Triton X-100, EDTA-free complete protease inhibitor (Roche Diagnostics)). Lysates were kept on ice for 20 min and centrifuged at 17,000 $\times$ g for 20 min to remove insoluble material. Supernatants were collected and protein concentration was determined using a BCA protein assay kit (Pierce).

## Click reaction and pull down

Lysates were thawed on ice. Proteins (100–300 µg) were taken and diluted to 1 mg/mL using the lysis buffer. A click mixture was prepared by adding reagents in the following order and by vortexing between the addition of each reagent: a capture reagent (stock solution 10 mM in water, final concentration 0.1 mM), $CuSO_4$ (stock solution 50 mM in water, final concentration 1 mM), TCEP (stock solution 50 mM in water, final concentration 1 mM), TBTA (stock solution 10 mM in DMSO, final concentration 0.1 mM). Following the addition of the click mixture the samples were vortexed (room temperature, 1 hr), and the reaction was stopped by addition of EDTA (final concentration 10 mM). Subsequently, proteins were precipitated (chloroform/methanol, 0.25:1, relative to the sample volume), the precipitates isolated by centrifugation (17,000 x g, 10 min), washed with methanol (1 $\times$ 400 µL) and air dried (10 min). The pellets were then resuspended (final concentration 1 mg/mL, PBS, 0.4% SDS) and the precipitation step was repeated to remove excess of the capture reagent. Next, samples were added to 15 µL of pre-washed (0.2% SDS in PBS (3 $\times$ 500 µL)) Dynabeads MyOne Streptavidin C1 (Invitrogen) and gently vortexed for 90 min. The supernatant was removed and the beads were washed with 0.2% SDS in PBS (3 $\times$ 500 µL).

## SDS-PAGE, in gel fluorescence and western blotting

Beads were supplemented with 2% SDS in PBS (20 µL) and 4x SLB (Invitrogen), boiled (95°C, 10 min), centrifuged (1000 x g, 2 min) and loaded on 10% or 4–20% SDS-PAGE gel (Bio-Rad). Following electrophoresis (60 min, 160V), gels were washed with water (3x). In-gel fluorescence was detected using a Pharos FX Plus Imager (Bio-Rad) and the protein loading was checked by Coomassie staining. For western blotting proteins were transferred (25 V, 1.3 A, 7 min) onto nitrocellulose membranes (Bio-Rad) using Bio-Rad Trans Blot Turbo transfer system. After a brief wash with PBS-T (PBS, 0.1% Tween-20) membranes were blocked (5% milk, TBS-T, 1 hr) and incubated with primary antibodies (5% milk, TBS-T, overnight, 4°C) at the following dilutions: rat anti-HA (1:1000; Roche Diagnostics), mouse anti-Myc (1:1000; Millipore), mouse anti-Ty1 (1:2000; Thermo Fisher), rabbit anti-Gra29 (1:1000; [*Young et al., 2020*]), rabbit anti-SFP1 (1:1000; [*Young et al., 2020*]), mouse anti-

Toxoplasma (1:1000; Abcam), mouse anti-CDPK1 (1:3000; Matt Bogyo Lab), rabbit anti-SAG1 (1:10,000; John Boothroyd Lab), rabbit anti-GAP45 (1:1000; Peter Bradley Lab), rabbit anti-*Tg*CAP (1:2000; [*Hunt et al., 2019*]), rabbit anti-MIC2 (1:500; Vernon Carruthers Lab), and mouse anti-MIC2 (6D10) (1:1000; Vernon Carruthers Lab). Following washing (TBS-T, 3x) membranes were incubated with IR dye-conjugated secondary antibodies from LI-COR Biosciences (1:10,000, 5% milk, TBS-T, 1 hr), and after a final washing step imaged on a LiCOR Odyssey imaging system (LI-COR Biosciences). In case of a biotin western blot, membranes were blocked with 3% BSA and incubated with Streptavidin-HRP (1:4000; Thermo Scientific) in 0.3% BSA, PBS-T for 1 hr. ECL western blotting Detection Reagent (GE Healthcare) was then used for chemiluminescence based imaging on a ChemiDoc MP Imaging System (Bio-Rad).

## Synthesis of capture reagents

TEV reagent: Solid phase synthesis took place on a CF peptide synthesizer (Intavis) using a Rink Amide LL resin (100 µmol; Merck) and $N(\alpha)$-Fmoc amino acids, including Fmoc-Lys(N$_3$)-OH (Fluorochem) and Fmoc-Gly-(Dmb)Gly-OH (Merck). HCTU was used as the coupling reagent with 5-fold excess of amino acids. Fmoc-Lys(Biotin)-OH (four eq; Merck) in 6 mL DMSO:NMP (1:1) was coupled manually after automated assembly of the rest of the chain. DIPEA (four eq) was added, followed by HOBt (1 M, four eq) in NMP. After 3 min DIC (four eq) was added, then after 30 min the solution was added to the resin and allowed to react overnight. The resin was washed with DCM and DMF prior to manual Fmoc removal and acetylation. The peptide was cleaved from the resin and protecting groups removed by addition of a cleavage solution (95% TFA, 2.5% H$_2$O, 2.5% TIS). After 2 hr, the resin was removed by filtration and peptides were precipitated with diethyl ether on ice. The peptide was isolated by centrifugation, then dissolved in H$_2$O and freeze dried overnight. After dissolving in methanol, portions of the peptide were purified on a C8 reverse phase HPLC column (Agilent PrepHT Zorbax 300 SB-C8, 21.2 × 250 mm, 7 m) using a linear solvent gradient of 13–50% MeCN (0.08% TFA) in H$_2$O (0.08% TFA) over 40 min at a flow rate of 8 mL/min. The peak fraction was analysed by LC–MS on an Agilent 1100 LC-MSD. The calculated molecular weight of the peptide was in agreement with the mass found. Calculated MW: 1804.08, actual mass: 1803.87.

Trypsin reagent: Solid phase synthesis took place on a CF peptide synthesizer (Intavis) using a Fmoc-PEG-Biotin NovaTag resin (100 µmol; Merck), 2-Azidoacetic acid (Fluorochem) and $N(\alpha)$-Fmoc amino acids, including Fmoc-Lys(MMT)-OH (Merck). HATU was used as the coupling reagent with 5-fold excess of amino acids. Following chain assembly, the MMT protecting group was removed from the peptidyl-resin by treatment with 1% TFA in DCM (10 mL for 2 min x 8) and the resin washed with DCM and DMF. Next 5-TAMRA (four eq; Anaspec) was dissolved in 1 mL DMSO:NMP (1:1). DIPEA (four eq) was added, followed by HOBt (1 M, four eq) in NMP. After 3 min DIC (four eq) was added, then after 30 min the solution was added to the resin and allowed to react overnight. After washing the resin with DMF and DCM, the peptide was cleaved from the resin and protecting groups removed by addition of a cleavage solution (95% TFA, 2.5% H$_2$O, 2.5% TIS). After 2 hr, the resin was removed by filtration and peptides were precipitated with diethyl ether on ice. The peptide was isolated by centrifugation, then dissolved in H$_2$O and freeze dried overnight. After dissolving in MeCN: H$_2$O (1:1), portions of the peptide were purified on a C8 reverse phase HPLC column (Agilent PrepHT Zorbax 300 SB-C8, 21.2 × 250 mm, 7 m) using a linear solvent gradient of 10–50% MeCN (0.08% TFA) in H$_2$O (0.08% TFA) over 40 min at a flow rate of 8 mL/min. The peak fraction was analysed by LC–MS on an Agilent 1100 LC-MSD. The calculated molecular weight of the peptide was in agreement with the mass found. Calculated MW: 1396.31, actual mass: 1395.60.

## Sample preparation for MS-based proteomics

Click reaction - Reagent **1** and **2**: lysates were thawed on ice and the click reaction was carried out with 1 mg of proteins at 2 mg/mL. Proteins were captured by adding a mixture of respective capture reagent (final concentration 0.1 mM), CuSO$_4$ (final concentration 1 mM), TCEP (final concentration 1 mM) and TBTA (final concentration 0.1 mM). The samples were vortex-mixed (room temperature, 1 hr) before the addition of EDTA (final concentration 10 mM), methanol (four volumes), chloroform (1 vol), and water (three volumes). The samples were vortex-mixed briefly, centrifuged (10,000 × g, 20 min) and the resulting pellets were either washed with methanol (four volumes) and dried (reagent **1**) or resuspended (at 2 mg/mL, 1% SDS in PBS) after which the precipitation step was repeated and

the resulting pellets washed with methanol (four volumes) and dried (reagent **2**). Reagent **3**: lysates were thawed on ice and the click reaction was carried out with 1 mg of proteins at 2 mg/mL. Proteins were captured by sequential addition of the capture reagent (final concentration 0.1 mM), TCEP (final concentration 1 mM), TBTA (stock in DMSO:t-Butanol 1:4, final concentration 0.1 mM) and CuSO$_4$ (final concentration 1 mM) with mixing between each step. The samples were incubated at room temperature for 1 hr before the addition of EDTA (final concentration 10 mM), methanol (four volumes), chloroform (1 vol), and water (three volumes). The samples were vortex-mixed briefly, centrifuged (10,000 × g, 20 min) and the resulting pellets were washed with methanol (four volumes) and dried. Subsequently, the dried pellets were resuspended in 2% SDS in PBS and, once completely dissolved, PBS was added (final concentration 0.8% SDS, 2 mg/mL). For samples treated with base, NaOH was added (final concentration 0.2 M, 1 hr) followed by neutralisation with equivalent amount of HCl. Base-treated and untreated samples were then diluted (1 mg/mL, 0.4% SDS, 1 mM DTT) before pull down.

Pull down, reduction and alkylation - NeutrAvidin agarose resin (Thermo Scientific) was washed with 0.2% SDS in PBS (3x). Typically, 50 µL of bead slurry was used for 1 mg of lysate. The samples were added to beads and the enrichment was carried out with gentle mixing (2 hr, room temperature). Following the removal of supernatants, the beads were sequentially washed with 1% SDS in PBS (3x), 4 M urea in PBS (2x) and 50 mM ammonium bicarbonate (3x). The samples were reduced (5 mM DTT, 56°C, 30 min) and cysteines alkylated (10 mM iodoacetamide, room temperature, 30 min) in the dark with washing the beads (2x, 50 mM ammonium bicarbonate) after each step.

Protein digestion - for samples processed with reagent **1** and **2** as well as for supernatants (proteomes) MS grade trypsin (Promega) was used at 1:1000 w/w protease:protein, and samples were incubated overnight at 37°C. For reagent **3** two digestion strategies were used. TEV I: beads were washed (2x) with water followed by TEV buffer (50 mM TrisHCl, 0.5 mM EDTA, 1 mM DTT, pH 8.0) and the TEV protease (50 units; Invitrogen) was added. Samples were incubated overnight at 30°C. Supernatant was then removed and beads washed with TEV buffer (1x, 50 µL). The wash fraction was combined with the supernatant and stored at 4°C. A fresh portion of TEV protease (20 units) was then added to beads which were incubated for additional 6 hr at 30°C. The supernatant and wash were combined with the first TEV elution. MS grade trypsin was subsequently added at 1:1000 w/w protease:protein, and samples were incubated overnight at 37°C. TEV II: samples were incubated overnight at 37°C with MS grade trypsin at 1:1000 w/w protease:protein. The supernatant was removed (fraction 1) and beads washed with water and TEV buffer (2x each). The TEV protease was then added (50 units) and beads incubated overnight at 30°C (fraction 2).

Stage tip - samples were desalted prior to LC-MS/MS using Empore C18 discs (3M). Each stage tip was packed with one C18 disc, conditioned with 100 µL of 100% methanol, followed by 200 µL of 1% TFA. The samples were loaded in 1% TFA, washed 3 times with 200 µL of 1% TFA and eluted with 50 µL of 50% acetonitrile, 5% TFA. Desalted peptides were vacuum dried in preparation for LC-MS/MS analysis.

## Chemical inhibition of *Tg*NMT

HFFs were infected with *Toxoplasma* and cultured for 16 hr. The medium was then replaced and intracellular parasites co-treated with 25 µM YnMyr and the indicated concentrations of IMP-1002 for 5 hr. Following PBS wash (2x) the cells were lysed on ice using the lysis buffer and further processed exactly as described above. The click reaction, pull down on NeutrAvidin beads and the MS sample prep were performed as described above for reagent **1** and **2**.

## LC-MS/MS

Samples were resuspended in 0.1% TFA and loaded on a 50 cm Easy Spray PepMap column (75 µm inner diameter, 2 µm particle size, Thermo Fisher Scientific) equipped with an integrated electrospray emitter. Reverse phase chromatography was performed using the RSLC nano U3000 (Thermo Fisher Scientific) with a binary buffer system (solvent A: 0.1% formic acid, 5% DMSO; solvent B: 80% acetonitrile, 0.1% formic acid, 5% DMSO) at a flow rate of 250 nL/min. Samples processed with reagent **1** were run on a linear gradient of 2–35% B in 90 min with a total run time of 120 min including column conditioning. Samples processed with reagents **2** and **3** were run on a linear gradient of 2–40% B or 2–55% B (TEV II myristoylated peptide fraction) in 155 min with a total run time of 180

min including column conditioning. The nanoLC was coupled to a Q Exactive mass spectrometer using an EasySpray nano source (both Thermo Fisher Scientific). The Q Exactive was operated in data-dependent mode, acquiring HCD MS/MS scans (R = 17,500) after an MS1 survey scan (R = 70,000) on the 10 most abundant ions using MS1 target of 1E6 ions, and MS2 target of 5E4 ions. The maximum ion injection time utilised for MS2 scans was 120 ms, the HCD normalised collision energy was set at 28 and the dynamic exclusion was set at 30 s. The peptide match and isotope exclusion functions were enabled. NMTi samples were run on a linear gradient of 2–20% B in 55 min, followed by 20–40% B in 35 min and 40–60% B in 5 min with a total run time of 120 min including column conditioning. The nanoLC was coupled to a Orbitrap Lumos mass spectrometer using an EasySpray nano source (both Thermo Fisher Scientific). The Orbitrap Lumos was operated in data-dependent mode (3 s cycle time), acquiring IT HCD MS/MS scans in rapid mode after an MS1 survey scan (R = 120, 000). The MS1 target was 4E5 ions whereas the MS2 target was 2E3 ions. The maximum ion injection time utilised for MS2 scans was 300 ms, the HCD normalised collision energy was set at 32 and the dynamic exclusion was set at 30 s.

## Data analysis

Acquired raw files were processed with MaxQuant, versions 1.5.0.25 and 1.5.2.8 (*Cox and Mann, 2008*). Peptides were identified from the MS/MS spectra searched against *Toxoplasma gondii* (combined TG1, ME49 and VEG proteomes, ToxoDB) and *Homo sapiens* (UniProt) proteomes using Andromeda (*Cox et al., 2011*) search engine. Cysteine carbamidomethylation was selected as a fixed modification and methionine oxidation was selected as a variable modification. The enzyme specificity was set to trypsin with a maximum of 2 missed cleavages. The precursor mass tolerance was set to 20 ppm for the first search (used for mass re-calibration) and to 4.5 ppm for the main search. The datasets were filtered on posterior error probability (PEP) to achieve a 1% false discovery rate on protein, peptide and site level. Other parameters were used as pre-set in the software. 'Unique and razor peptides' mode was selected to allow identification and quantification of proteins in groups (razor peptides are uniquely assigned to protein groups and not to individual proteins). Label-free quantification (LFQ) in MaxQuant was performed using a built-in label-free quantification algorithm (*Cox and Mann, 2008*) enabling the 'Match between runs' option (time window 0.7 min) within replicates. Each experiment comprised of replicates treated with YnMyr and the same number of replicates treated with Myr control or NMTi. The LFQ is based on intensities of proteins calculated by MaxQuant from peak intensities and based on the ion currents carried by peptides whose sequences match a specific protein or a protein group to provide an approximation of abundance.

Myristoylated peptide search in MaxQuant was performed as described above applying the following variable modifications: cysteine carbamidomethylation, +463.2907 (reagent **2**) and +491.3220 (reagent **3**) at any peptide N-terminus and cysteine residues. In addition, the minimum peptide length was reduced to six amino acids and the 'Match between runs' option was disabled. MaxQuant utilises a scoring algorithm when matching experimental MS/MS spectra with a library of theoretical spectra generated from the in silico digestion of proteins within databases selected for the search. The algorithm is used to evaluate the quality of peptide-spectrum matches (PSMs). To each PSM, MaxQuant also attributes a delta score, which is a difference between scores associated with the match to the best peptide candidate and the second best match within the database. The higher the score and the delta score, the more reliable the identification. In order to reduce a possibility for a false peptide sequence assignment even further, we applied relatively high delta score thresholds (20 vs 6 pre-set as default) for all myristoylated peptides in our analysis. MaxQuant output files were processed with Perseus, version 1.5.0.9 (*Tyanova et al., 2016*) as described in the Results section and in *Supplementary files 1–3*. The mass spectrometry proteomics data have been deposited to the ProteomeXchange Consortium via the PRIDE (*Perez-Riverol et al., 2019*) partner repository with the dataset identifier PXD019677.

## Homology modelling of *Tg*NMT onto *Pv*NMT crystal structure

The model of *Tg*NMT was generated using SWISS-MODEL (*Waterhouse et al., 2018*), and aligned with a crystal structure of *Pv*NMT bound to NMT inhibitor IMP-1002 (PDB: 6MB1). The structural image was generated using PyMOL (Schrodinger LLC (2010), The PyMOL Molecular Graphics System, Version 1.3r1).

## MIC7 expression in tachyzoites and bradyzoites

HFF monolayers were infected with Pru Δhxgprt parasites in triplicate. For tachyzoite samples an MOI of 1 was used for a 27 hr infection. For bradyzoite samples monolayers were infected at an MOI of 0.8 for 3.5 hr, washed and grown in switch conditions (RPMI, 1% FBS, pH 8.1, ambient $CO_2$) for 3 days. Triplicate samples were lysed in 2 mL ice cold lysis buffer (50 mM Tris-HCl, 75 mM NaCl, 8 M urea, pH 8.2), supplemented with protease (Roche Diagnostics) and phosphatase (Phos Stop, Roche Diagnostics) inhibitors. Lysis was followed by sonication to reduce sample viscosity (30% duty cycle, $3 \times 30$ s bursts, on ice). Protein concentration was measured using a BCA protein assay kit (Pierce). Lysates (1 mg per condition) were subsequently processed for mass spectrometry as described (*Young et al., 2020*) and data analysis performed as explained in *Supplementary file 5*. For full dataset please see *Young et al., 2020* and PXD019729.

## Depletion of *Mic7*

Parasites were allowed to invade HFFs for 2 hr and then treated with 50 nM rapamycin (Sigma-Aldrich) or an equivalent volume of vehicle (DMSO) for 4 hr. The medium was then replaced and the parasites allowed to grow for at least 24 hr prior to PCR and western blot analysis.

Parasite preparation for large scale invasion assay/live microscopy/gliding:

HFF monolayers in T25 flasks were infected in culture conditions (37°C and 5% $CO_2$) with recently egressed tachyzoites to achieve a one to two-per-cell parasite infection. Non-internalized parasites were removed with PBS, and the infected monolayers were cultivated for about 2 hr in complete culture medium (DMEM supplemented with 10% FCS, 10 mM HEPES, 100 units/mL penicillin, and 100 mg/mL streptomycin). After 2 hr incubation the parasites were treated with 50 nM rapamycin or vehicle. Following a 14 hr incubation the medium was replaced with the complete medium and tachyzoites were used within 2 to 5 hr post-egress.

## Plaque assays

Parasites were harvested by syringe lysis, counted, and 400 parasites were seeded on confluent HFF monolayers grown in 24-well plates (Falcon). Parasites were allowed to invade overnight prior to treatment with 50 nM rapamycin or vehicle (DMSO) for 4 hr. Following medium replacement to standard culture medium, plaques were allowed to form for 5 days. iKO MIC7 line: Parasites were harvested by syringe lysis, counted, and 100 parasites were seeded on confluent HFF monolayers grown in 24-well plates (Falcon). Parasites were allowed to invade overnight prior to treatment with 50 nM rapamycin or vehicle (DMSO) for 4 hr. Following medium replacement to standard culture medium, plaques were allowed to form for 7 days. NMTi: Parasites were harvested by syringe lysis, counted, and 200 parasites were seeded on confluent HFF monolayers grown in the presence of IMP-1002 for 5 days.

Plaque formation was assessed by inspecting the methanol fixed and 0.1% crystal violet stained HFF monolayers.

## Immunofluorescence analysis

Parasite-infected HFF monolayers grown on glass coverslips were fixed with 3% formaldehyde for 15 min prior to washing with PBS. Fixed cells were then permeabilised (0.2% Triton X-100/PBS, 10 min), blocked (3% BSA/PBS, 1 hr) and stained for 1 hr with primary antibodies at the following dilutions: rat anti-HA (1:1000; Roche), mouse anti-Myc (1:1000; Millipore), mouse anti-Ty1 (1:500; Thermo Fisher), rabbit anti-MIC2 (1:5000; Vernon Carruthers Lab). Labelled proteins were visualised with Alexa Fluor-conjugated secondary goat antibodies (1:2000, 1 hr; Life Technologies). Nuclei were visualised with the DNA stain DAPI (5 µg/mL; Sigma) supplemented with the secondary antibody. Stained coverslips were mounted on glass slides with Slowfade (Life Technologies) and imaged on a Nikon Eclipse Ti-U inverted fluorescent microscope using 100x oil objective. Images were analysed using Nikon NIS-Elements imaging software.

## Invasion assay

Parasites were treated with 50 nM rapamycin for 4 hr and after replacing the medium allowed to grow for 24 hr. Red/green invasion assays were then performed. Parasites were lysed in an invasion non-permissive buffer, Endo buffer (44.7 mM $K_2SO_4$, 10 mM $MgSO_4$, 106 mM sucrose, 5 mM

glucose, 20 mM Tris–H$_2$SO$_4$, 3.5 mg/mL BSA, pH 8.2). 250 µL of 8E5 parasites/mL in Endo buffer were added to each well of a 24-well flat-bottom plate (Falcon) containing a coverslip with a confluent HFF monolayer. The plates were spun at 129 x g for 1 min at 37°C to deposit parasites onto the monolayer. The Endo buffer was gently removed and replaced with invasion permissive medium (1% FBS/DMEM). Parasites were allowed to invade for 15 min at 37°C, after which the monolayer was gently washed with PBS and fixed with 3% formaldehyde for 15 min at room temperature. Extracellular (attached) parasites were stained with mouse anti-*Toxoplasma* (TP3) (1:1000; Abcam) and goat anti-mouse Alexa Fluor 488 before permeabilisation (0.2% Triton X-100/PBS) and detection of intracellular (invaded) parasites with rabbit anti-*Tg*CAP (1:2000; [*Hunt et al., 2019*]) and goat anti-rabbit Alexa Fluor 594. For each replicate, at least five random fields were imaged with a 40x objective. Three independent experiments were performed in duplicate. The number of intracellular (594+/ 488-) and extracellular (594+/488+) parasites was determined by counting, in a blinded fashion, at least 275 parasites per strain. The parasite counts in the MIC7 iKO and cMut lines were normalised to the cWT, and results were statistically tested with a one-way ANOVA with Dunnett's multiple comparison test in GraphPad Prism 8. The data are presented as mean ± SD. For estimation of the parasite attachment efficiency, the number of all (594+) parasites was used and the results were statistically tested as above.

## Large scale automated invasion assay

Cell Invasion - HFFs were seeded at a density of 2E4 cells per well into 96-well plate and cultivated in complete medium at 37°C and 5% CO$_2$ for 24 hr to allow for sub-confluence. 5E6 to E7 parasites were collected upon spontaneous egress from synchronously infected HFF monolayers. The supernatant was centrifuged at higher speed (900 x g, 7 min) to collect parasites that were gently suspended in 2 mL of complete medium before counting. 2.5E5 parasites were added to each well. To synchronize invasion, the 96-well plate was centrifuged (300 x g, 3 min) and incubated for 30 min at 37°C and 5% CO$_2$. After gentle aspiration, invasion was stopped by addition of 3.2% paraformaldehyde (PFA) in PBS, pH 7.5 (20 min).

Parasite staining - immunostaining was performed first under conditions that did not permeabilise the HFF cells to allow discriminating between the extracellular and intracellular parasites. Fixed cell samples were incubated in 2%BSA/PBS as a blocking buffer (BB) for 20 min. Extracellular tachyzoites were selectively stained using mouse anti-*Tg*SAG1(TP3) (1 mg/mL stock, 1:600, 40 min; Novocastra) followed by Alexa Fluor 488-conjugated highly cross-adsorbed (HCA) anti-mouse antibody (2 mg/mL stock, 1:800, 1 hr). The excess of reagents was washed off with PBS and cells were permeabilised (0.2% Triton X-100/PBS, 10 min) prior to incubation in BB and the second staining step using anti-*Tg*SAG1 (1 mg/mL stock, 1:600, 1 hr) but followed by Alexa Fluor 594-conjugated HCA anti-mouse antibody (2 mg /ml stock, 1:800, 1 hr). Cell nuclei were stained with 500 nM DAPI and the 96 well plates were automatically scanned to quantify the average number of cells per well. The nuclei of parasites are detected by blue fluorescence whereas the intracellular tachyzoites by red fluorescence and the extracellular ones by yellow fluorescence (as the result of green and red fluorescence).

Quantification - samples were automatically scanned at a magnification of 20x under an Olympus ScanR automated inverted microscope. Images were acquired for five wells per parasite strain for each invasion assay, with 16 randomly scanned fields per well and further processed with ScanR software. ScanRAnalysis includes algorithms to provide automated cell nuclei segmentation following signal-to-noise ratio optimisation and accurate cell surface mask definition. To identify intracellular (red) over extracellular (yellow) parasites, image subtraction from each channel was automatically obtained. Data collected allowed determining the total number of intracellular tachyzoites over the total number of host cells for each well. Three independent assays were carried out, and data were statistically analysed using a two-tailed Student's t-test in GraphPad Prism 8. The data are presented as mean ± SD.

## Invasion video microscopy

Preparation of human cells - HFF and Human Bone OsteoSarcoma cells (U2OS) that stably expressed the GFP-GPI plasma membrane reporter, were seeded at a density of 3E5 cells per 18 mm glass coverslip, previously coated with poly-L-lysine (50 µg/mL). Cells were cultivated in complete medium at

37°C and 5% $CO_2$ for 24 hr to allow for 80% confluence. Coverslips were placed in Chamlide chambers (LCI Corp.) and covered with a minimal volume (i.e. 100 µL) of motility buffer (see below).

Preparation of parasites – 2E5 to 4E5 parasites were typically collected upon spontaneous egress from synchronously infected HFF monolayers. 150 µL of this suspension were mixed with 5 mL of Hanks' Balanced Salt Solution (HBSS) supplemented with 0.2% FCS. After centrifugation (900 x g, 7 min), parasites were resuspended in 200 µL of motility buffer (HBSS supplemented with 1% FCS and 0.5 mM $CaCl_2$ to reach about 1.6 mM $CaCl_2$ final). Typically, 30 to 40 µL of the suspension were added to the cells on the coverslip immobilized in the chamber, to avoid parasite overcrowding during recording.

Video recording of the tachyzoite behaviour - the recording chamber that accommodates the coverslips was installed on an Eclipse Ti inverted confocal microscope (Nikon) to perform time-lapse video microscopy, with a temperature and $CO_2$-controlled stage (LCI Corp.). The microscope was also equipped with a CMOS camera and a CSU X1 spinning disk (Yokogawa, Roper Scientific). The microscope was piloted using MetaMorph software (Universal Imaging Corporation, Roper Scientific). Similar parameters for image acquisition were used throughout each independent experiment. Time of invasion was estimated for each tachyzoite using MetaMorph time scale between the moment of contact between parasite apex (i.e. conoid) and host cell membrane until the tachyzoite has fully passed through the cell-zoite junction. Time of failed invasion was quantified using the same software, once again between the time of apical contact to those of body withdrawal and detachment, or the moment the tachyzoite did not perform any movement.

## Gliding assays

Freshly egressed parasites of each genetic background were prepared as for the video recording assays. About 2 to 4E5 tachyzoites in 300 µL of motility buffer were deposited on 12 mm glass coverslip, previously coated with poly-L-lysine (as above) and placed in a 24 well plate. Parasites were gently centrifuged (200 x g, 3 min) to ensure rapid contact with the coverslip and then allowed to glide for 10–15 min at 37°C and 5% $CO_2$. Motile activity was checked under microscope after a few first minutes. At the end of this period, the samples were fixed after gentle aspiration of the liquid by the addition of 3.2% PFA in PHEM pH 7.5 (20 min). Trails left by gliding parasites and parasite surface were stained after a blocking step (2% BSA/PBS, 30 min) with mouse anti-*Tg*SAG1 antibody (1 mg/mL stock, 1:600, 2 hr) and Alexa Fluor 488-conjugated HCA anti-mouse antibody (2 mg/mL stock, 1:800, 2 hr). Cell nuclei were stained with 500 nM DAPI and mounted in Mowiol. Images of trails and tachyzoites were captured under the fluorescent ApoTome two microscope (Zeiss) using appropriate set of filters, the Zen software (Zeiss) and a z step of 0.3 µm. Image stacks were further processed with FIJI (*Schindelin et al., 2012*) and Photoshop.

## Immunofluorescence of successful and failed invasions

MIC7 iKO parasites were treated with DMSO or rapamycin and were prepared as described for the video recording assays. Approximately 8E5 tachyzoites in 300 µL of motility buffer were deposited on 12 mm glass coverslip previously coated with poly-L-lysine (as above) and placed in a 24 well plate. Tachyzoites were gently centrifuged (300 x g, 3 min) to ensure rapid contact with the host cell (non-fluorescent HFF and GFP-GPI expressing U2OS cells) and allowed to invade for 2 to 4 min periods at 37°C and 5% $CO_2$. Samples were immediately fixed in 3.2% PFA in PHEM pH 7.5 (20 min) prior to be processed for IFA. Blocking step and anti-*Tg*SAG1 staining were performed as for the gliding assay except that the incubation with SAG1 primary antibody and Alexa Fluor 633-conjugated secondary antibody was reduced to 30 min. After SAG1 staining, the HFF-tachyzoite samples were permeabilised with (0.2% Triton X-100/PBS, 5 min) prior to a second step of blocking. MIC7 labelling was performed using rabbit anti-HA (clone C29F4), (1:800, 2 hr; Cell Signaling) followed by Alexa Fluor 488-conjugated HCA anti-rabbit antibody (2 mg/mL stock, 1:800, 2 hr).

## MIC7 dimerisation

HFF monolayers infected with parasites from the DiCre, $^{Myc}$cWT and $^{Myc}$cMut lines were washed with cold PBS and lysed in IP buffer (50 mM Tris, 150 mM NaCl, 0.2% Triton-X100, pH7.5) supplemented with protease inhibitors (Roche Diagnostics) for 30 min on ice. The lysates were then centrifuged (5000 x g, 20 min, 4°C), the supernatants collected and incubated with 20 µL of α-HA-

conjugated agarose beads (Millipore) on a rotating wheel at 4°C. After 3 hr the supernatant was removed and beads washed 3x with IP buffer. Protein elution from beads was performed with SDS sample loading buffer and boiling at 95°C for 10 min. Input, IP and supernatant samples for each tested parasite line were then analysed by SDS-PAGE and western blotting.

### Microneme secretion assay

Parasites from MIC7 iKO line were treated with DMSO or rapamycin (50 nM, 4 hr). After 24 hr incubation parasites were syringe lysed in DMEM at room temperature and collected by centrifugation (800 x g, 4°C, 10 min). Pellets were resuspended in Ringer's buffer (155 mM NaCl, 3 mM KCl, 2 mM CaCl$_2$, 1 mM MgCl$_2$, 3 mM NaH$_2$PO$_4$, 10 mM HEPES, 10 mM glucose) supplemented with BIPPO (50 μM) or vehicle and microneme secretion was induced at 37°C for 20 min. Following this incubation step the parasites were placed on ice and pelleted (1000 x g, 5 min, 4°C). The pellet was kept on ice while the supernatant was re-pelleted (2000 x g, 5 min, 4°C). The final supernatant, containing the excreted secreted antigens, and pellet fractions were resuspended in sample loading buffer prior to SDS-PAGE and western blotting.

### Shedding assays

Shedding tests during egress for $^{Myc}$cWT and $^{Myc}$cMut lines were performed exactly as described in microneme secretion assay.

To test for MIC7 shedding upon invasion, parasites from the iKO, $^{Myc}$cWT and $^{Myc}$cMut lines were treated with DMSO or rapamycin (50 nM, 4 hr). After 24 hr incubation parasites were syringe lysed in cold DMEM and spun (300 x g, 3 min, 4°C) onto PBS washed HFF monolayers in a 6-well plate (Falcon). The plate was then incubated at 37°C to facilitate invasion. After 1 hr the plate was placed on ice, the supernatant was gently aspirated off and spun down (700 x g, 10 min, 4°C) to remove any aspirated parasites. Proteins were precipitated out by the addition of cold trichloroacetic acid (10% v/v) on ice (30 min). Samples were centrifuged (17,000 x g, 20 min, 4°C), washed with 300 μL of cold acetone and air dried. The infected monolayers were scraped in 0.5 mL cold PBS and collected by centrifugation (17,000 x g, 20 min, 4°C). Both pellet and supernatant samples were resuspended in sample loading buffer prior to SDS-PAGE and western blotting.

### Proteinase K protection assay

Parasites from $^{Myc}$cWT and $^{Myc}$cMut lines were treated with rapamycin (50 nM, 4 hr) followed by YnMyr (25 μM, 16 hr). Parasites were then syringe lysed in DMEM at room temperature and collected by centrifugation (800 x g, 10 min, 4°C). Pellets were resuspended in 1.7 mL cold SoTE buffer (0.6 M sorbitol, 20 mM Tris–HCl (pH 7.5), and 2 mM EDTA) and split into three tubes (0.5 mL each) per tested parasite line. Tubes 2 and 3 were permeabilised with 0.01% cold Digitonin (Sigma-Aldrich) in SoTE. Samples were carefully mixed by inversion and incubated on ice (10 min) prior to centrifugation (1000 x g, 10 min, 4°C). Supernatant was discarded. Pellets were resuspended in 0.5 mL cold SoTE and 8 μg of Proteinase K (Sigma-Aldrich) were added to tube 3. All tubes were gently inverted and incubated on ice (30 min). Proteinase K was inactivated by addition of ice cold trichloroacetic acid to a final concentration of 10% v/v on ice (30 min). Samples were centrifuged (17,000 x g, 20 min, 4°C), washed with 300 μL of cold acetone and air dried prior to SDS-PAGE and western blotting.

### Rhoptry secretion assay

Parasites from iKO, $^{Myc}$cWT and $^{Myc}$cMut lines were treated with 50 nM rapamycin or an equivalent volume of vehicle (DMSO) for 4 hr after which the medium was replaced and the parasites allowed to grow for 24 hr. Parasites were harvested by syringe lysis, counted, and treated with 1 μM Cytochalasin D (Sigma) for 10 min at room temperature. 500,000 parasites from each condition were seeded onto confluent HFF monolayers grown in chambered coverslip slides (ibidi) and allowed to settle for 10 min on ice. The slides were spun down (250 x g, 1 min, 4°C) then transferred to a 37°C water bath for 20 min to initiate rhoptry secretion. The chambers were washed 3x with PBS then fixed with ice-cold methanol at −20°C for 8 min and washed 3x with PBS. Fixed cells were permeabilised with 0.1% Triton X-100 in PBS for 15 min then blocked with 3% BSA in PBS for 1 hr. Cells were then incubated with rabbit anti-phospho-Stat6 (1:600; Cell Signaling) and mouse anti-*Toxoplasma*

(TP3) (1:1000; Abcam) primary antibodies for 1 hr. After 3x washes with PBS, cells were incubated with goat anti-rabbit Alexa Fluor 594 (1:2000; Life Technologies) and goat anti-mouse Alexa Fluor 488 (1:2,000; Life Technologies) secondary antibodies and 5 µg/mL DAPI (Sigma) for 1 hr followed by 3x washes with PBS. Images were obtained using a Nikon Eclipse Ti-U inverted fluorescent microscope using a 20x objective and analysed using FIJI software. $\geq$5 fields of view per condition were analysed in three independent experiments. The number of pSTAT6 positive HFFs was normalised to the total number of HFFs, and results were statistically tested with a two-way ANOVA with Sidak's multiple comparison test in GraphPad Prism 8. The data are presented as mean ± SD.

## Acknowledgements

We would like to thank Peter Bradley, Matt Bogyo, Vern Carruthers, and John Boothroyd for sharing reagents as well as the teams from EupathDB and ToxoDB for their valuable contributions to the community. We also thank members of the Proteomics and Peptide Synthesis Science Technology platforms at The Francis Crick Institute. This work was supported by funding from The Francis Crick Institute (https://www.crick.ac.uk/), which receives its core funding from Cancer Research UK (FC001189; FC001999), the UK Medical Research Council (FC001189; FC001999) and the Wellcome Trust (FC001189; FC001999) as well as the National Institute of Health grant (NIH-R01AI123457). SM was supported by the Leverhulme Trust (RPG-2018–107), IT by the internal grant from the Prevention and Therapy of Chronic Diseases call at the IAB, and EWT by the Cancer Research UK Programme Foundation Award C29637/A20183.

## Additional information

### Competing interests

Edward W Tate: EWT is a founder, shareholder and Director of Myricx Pharma Ltd. The other authors declare that no competing interests exist.

### Funding

| Funder | Grant reference number | Author |
|---|---|---|
| Francis Crick Institute | FC001189 | Malgorzata Broncel<br>Caia Dominicus<br>Stephanie D Nofal<br>Alex Hunt<br>Bethan A Wallbank<br>Joanna C Young<br>Moritz Treeck |
| NIH Office of the Director | R01AI123457 | Malgorzata Broncel<br>Caia Dominicus<br>Moritz Treeck |
| Leverhulme Trust | RPG-2018-107 | Stephen Matthews |
| Cancer Research UK | C29637/A20183 | Edward J Bartlett<br>Edward W Tate |
| Francis Crick Institute | FC001999 | Stefania Federico<br>Joanna C Young<br>Moritz Treeck |
| Institute for Advanced Biosciences | Prevention and Therapy of Chronic Diseases grant | Isabelle Tardieux |

The funders had no role in study design, data collection and interpretation, or the decision to submit the work for publication.

### Author contributions

Malgorzata Broncel, Conceptualisation, Data curation, Validation, Investigation, Visualisation, Methodology, Writing - original draft, Writing - review and editing; Caia Dominicus, Investigation, Methodology, Writing - review and editing; Luis Vigetti, Stephanie D Nofal, Investigation, Visualisation,

Writing - review and editing; Edward J Bartlett, Alex Hunt, Investigation, Visualisation; Bastien Touquet, Bethan A Wallbank, Investigation; Stefania Federico, Resources; Stephen Matthews, Joanna C Young, Investigation, Writing - review and editing; Edward W Tate, Resources, Writing - review and editing; Isabelle Tardieux, Visualisation, Writing - review and editing; Moritz Treeck, Conceptualisation, Supervision, Funding acquisition, Methodology, Writing - original draft, Writing - review and editing

### Author ORCIDs

Malgorzata Broncel https://orcid.org/0000-0003-2991-3500
Luis Vigetti https://orcid.org/0000-0001-9733-2770
Stephanie D Nofal https://orcid.org/0000-0003-1415-3369
Alex Hunt http://orcid.org/0000-0001-7431-7156
Bethan A Wallbank https://orcid.org/0000-0002-6432-2135
Stephen Matthews http://orcid.org/0000-0003-0676-0927
Isabelle Tardieux https://orcid.org/0000-0002-5677-7463
Moritz Treeck https://orcid.org/0000-0002-9727-6657

### Decision letter and Author response

Decision letter https://doi.org/10.7554/eLife.57861.sa1
Author response https://doi.org/10.7554/eLife.57861.sa2

## Additional files

### Supplementary files

• Supplementary file 1. related to *Figure 2*. Identification of base-dependent YnMyr enrichment in *T. gondii.* Sheet 1: *Toxoplasma* proteins with YnMyr intensities quantified irrespective of base treatment. Sheet 2: Proteins with base-sensitive enrichment. Sheet 3: MG proteins insensitive to base treatment and robustly enriched in a YnMyr-dependent manner with $N_3$-biotin reagent (**1**). Sheet 4: Analysis of proteomes (supernatants post enrichment).

• Supplementary file 2. related to *Figure 2*. Identification of myristoylated proteins and myristoylated peptides in *T. gondii*. Sheet 1: *Toxoplasma* proteins bearing the MG motif. Sheet 2: Substrates significantly enriched with Trypsin reagent (**2**). Sheet 3: Substrates selected based on fold change in YnMyr/Myr enrichment with TEV reagent (**3**). Sheet 4: Myristoylated peptides found with Trypsin reagent (**2**). Sheet 5: Myristoylated peptides found with TEV reagent (**3**). Sheet 6: Human proteins bearing the MG motif. Sheet 7: Human substrates significantly enriched with Trypsin and TEV reagents.

• Supplementary file 3. related to *Figure 3*. Chemical inhibition of *Tg*NMT. Sheet 1: Response of YnMyr enriched *Toxoplasma* proteins to NMTi. Sheet 2: NMTi does not significantly affect *Toxoplasma* proteome. Sheet 3: Response of base-sensitive *Toxoplasma* proteins to NMTi. Sheet 4: Response of YnMyr enriched Human proteins to NMTi. Sheet 5: NMTi does not significantly affect Human proteome.

• Supplementary file 4. related to *Figure 4*. Myristoylated proteome of *T. gondii*. Sheet 1: Substrate list and annotation. Sheet 2: Myristoylated proteins in *P. falciparum* and their orthologues in *Toxoplasma.* Sheets 3–9: Substrate orthologues in selected Apicomplexans.

• Supplementary file 5. related to *Figure 5*. MIC7 expression in tachyzoites and bradyzoites.

• Supplementary file 6. Primers used for plasmid and parasite lines generation.

• Transparent reporting form

### Data availability

All data generated or analysed during this study are included in the manuscript and supporting files. Source data files have been provided for Figures 5, 6 and 7. Source data for mass spectrometry proteomics results can be found in Supplementary files 1-4. The mass spectrometry proteomics data

have been deposited to the ProteomeXchange Consortium via the PRIDE (Perez-Riverol et al., 2019) partner repository with the dataset identifier PXD019677.

The following dataset was generated:

| Author(s) | Year | Dataset title | Dataset URL | Database and Identifier |
|---|---|---|---|---|
| Broncel M, Dominicus C, Vigetti L, Nofal SD, Bartlett EJ, Touquet B, Hunt A, Wallbank BA, Federico S, Matthews S, Young JC, Tate EW, Tardieux I, Treeck M | 2020 | Global profiling of myristoylation in *Toxoplasma gondii*. | http://proteomecentral. proteomexchange.org/ cgi/GetDataset?ID= PXD019677 | ProteomeXchange, PXD019677 |

The following previously published datasets were used:

| Author(s) | Year | Dataset title | Dataset URL | Database and Identifier |
|---|---|---|---|---|
| Koreny L, Ke H, Butterworth S, Crook OM, Lassadi I, Gupta V, Tromer E, Mourier T, Stevens TJ, Breckels LM, Pain A, Lilley KS, Waller RF | 2020 | Hyper LOPIT Global mapping of protein subcellular location. | https://toxodb.org/toxo/ app/record/dataset/DS_ eda79f81b5 | ToxoDB, DS_ eda79f81b5 |
| Young J, Broncel M, Teague H, Russell M, McGovern O, Renshaw M, Frith D, Snijders B, Collinson L, Carruthers V, Ewald S, Treeck M | 2020 | Differential protein phosphorylation during stage conversion in *Toxoplasma gondii* | http://proteomecentral. proteomexchange.org/ cgi/GetDataset?ID= PXD019729 | ProteomeXchange, PXD019729 |

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
