## [Decision Letter]

**Acceptance summary:**

The paper uses chemoproteomic methods to globally identify myristoylated proteins-including many modified peptides in the obligate intracellular parasite *Toxoplasma gondii*. This well performed myristylome is viewed as important and reliable resource for the scientific community. The authors also report the unexpected finding of a secreted protein implicated in invasion that is myristoylated at the N-terminus, implying an enigmatic trafficking mechanism.

**Decision letter after peer review:**

[Editors’ note: the authors submitted for reconsideration following the decision after peer review. What follows is the decision letter after the first round of review.]

Thank you for submitting your work entitled "Global profiling of myristoylation in *Toxoplasma gondii* reveals key roles for lipidation in CDPK1 and MIC7 function" for consideration by *eLife*. Your article has been reviewed by a Senior Editor, a Reviewing Editor, and three reviewers. The reviewers have opted to remain anonymous.

Our decision has been reached after consultation between the reviewers. Based on these discussions and the individual reviews below, we regret to inform you that your work will not be considered further for publication in *eLife*.

As you can see below, the three reviewers have raised some serious issues that would require substantial work and time to be addressed. Indeed any experiments that would mechanistically explore the role of myristoylation for MIC7 or CDPK1 would take considerably longer than two months, and the same holds true for the curation and interpretation of the myristoylome as well as providing data with NMT inhibitors for recovery of true Myred proteins. These weaknesses preclude consideration for publication.

However, if you feel that you can address all the concerns and would like to try a new resubmission after a substantial revision of the article, we will be prepared to reconsider it. Although the article will be treated as a new submission, we will do our best to send it to the same reviewers. If you choose this course of action please submit a separate cover letter detailing the changes you have made.

Reviewer #1:

The paper by Broncel et al., reports the identification by using targeted MS, of 65 Myristoyled proteins in *Toxoplasma gondii*, including CDPK1 and MIC7 (a predicted type-I-transmenbrane protein. The authors provided data, suggesting that Myristoylation of MIC7 is important in host cell invasion.

This is a very interesting paper in the field of lipid modifications in the obligate protozoan parasite *Toxoplasma gondii* and I particularly appreciated the strategy(ies) used to depict these tricky modifications on proteins. Nonetheless, the specificity of the MIC7Myristoylation effect claimed in the paper, which is not given a mechanistic explanation except for the possible involvement in host cell invasion, should be thoroughly re-considered to make the paper really convincing.

The experiments appear accurately preformed and well presented but a number of points required clarification.

Major concerns:

1) As the authors reported and already known, YnMyr can be non-specifically incorporated on amino acids differently than N-terminal glycines given the high level of non-myristoylation dependent background. To identify the YnMyr incorporation into N-terminal glycines, the commonly accepted method is the use of NMT inhibitors.

I understand the authors' method to distinguish between incorporation of YnMyr in N-Gly or GPI anchors, but to be convincing and to validate the base treatment the authors should also show data with NMT inhibitors for recovery of true Myred proteins.

2) Authors should report the list of 206 human proteins claimed to bear the MG myristoylation motif (subsection “Identification of the myristoylated proteome in *T. gondii”*) and compare the retrieved proteins with the recent reported complete list of human Myred proteins (Castrec et al., 2018) to validate their approach. This list is missing for the moment.

3) Alternative experiments concerning the cellular localization of CDPK1 will help better to understand the function of lipidation of this protein in *Toxoplasma gondii*. The data presented are not convincing.

4) As reported above, no mechanistic inside is reported to explain how myristoylation of MIC7 plays a role in the invasion of the host cells. This is a very important aspect of the work that should be further investigated.

5) Of all myristoylated proteins identified is not clear why the authors decided to focus on the two specific proteins (CDPK1 and MIC7). Although I can understand the characterization of MIC7, I'm a bit sceptic about the advances brought by CDPK1 data (already known to undergo myristoylation in most organism).

Reviewer #2:

Broncel et al., uses chemoproteomic techniques to identify myristoylated proteins in *Toxoplasma* and then validate two targets and derive functional information about the role of this modification during the lytic cycle. Myristoylation is a non-reversible modification (as compared to palmitoylation) and can occur either by itself or in partnership with nearby palmitoylation. Indeed, it is dogma that myristoylation is required for subsequent palmitoylation, at least in *Toxoplasma*. This work proceeds work done on the identification of palmitoylated proteins in both *Toxoplasma* and Plasmodium as well as the identification of myristoylated proteins in *P. falciparum* and investigations on its importance. Further, proteins that are both myristoylated and palmitoylated have been predicted, their localisation and function of these sites, in combination, investigated. The substrates that Broncel et al., identify here include these dually acylated proteins, validating their approach and also include many novel molecules. Broncel also duly acknowledge all this previous work.

Whilst this reviewer is not a chemical biologist, the approach taken to identify myristoylated proteins seems robust, well controlled and has been well validated. This dataset will be useful for others that are interested in acylation in *Toxoplasma* or determining whether their protein of interest is myristoylated. Further, this dataset also provides potential information on whether *Toxoplasma* proteins have a GPI anchor.

Presumably, because of the extensive work been done on dual acylation (myristoylation and palmitoylation together) the authors chose to study proteins that they suspect (but not proven) to be only myristoylated and not palmitoylated. They were able to show that the myristoylation site of both CDPK1 and MIC7 are functionally important, albeit limited.

The main issues I have with this paper is one of novelty. Whilst this work is robust, well validated and its functional relevance investigated, none of this is surprising. The real interesting question is, and where the field is at, is understanding what myristoylation is doing over and above membrane affinity and what relevance does this have to parasite biology? Again, it is unsurprising that loss of myristoylation leads to changes in solubility (as a surrogate for localisation), but the real interest is what other function might this modification have? This could be particularly interesting with MIC7 where it appears not to have an SP, and myristoylation not required for apparent correct localisation. What about is whether myristoylation is required for surface display or secretion? This might review novel biology of this lipid modification.

Reviewer #3:

Acylation has been shown to influence eukaryotic protein function in a variety of contexts. In *Toxoplasma gondii* several studies have shown acylation to be important for specific proteins, and global studies have been undertaken to describe the complement of palmitoylated proteins. However, the full extent of myristoylation-a modification often required to license palmitoylation, but also important in its own right-has not been systematically examined in the context of the *T. gondii* proteome. The present study is an important resource, addressing this gap in our understanding. Through several complementary mass-spectrometry experiments, the authors identify novel and known myristoylated proteins. For two of the identified proteins, CDPK1 and MIC7, the authors use complementation of knockouts to demonstrate the functional importance of myristoylation. Although, the precise role of myristoylation is not uncovered for either candidate, the evidence supporting a role for the modification is strong. The major concerns listed below therefore fall into two main categories: (1) curating the results and interpretation of the myristoylome to ensure its utility as a resource, and (2) ensuring the rigor of test cases.

1) In reference to Figure 1—figure supplement 1A, the statement "without any detectable background" is not supported by the visible background in the figure. The authors later also state that "this additional layer of confidence in target identification is especially important given the high level of non-myristoylation dependent background reported for metabolic tagging with YnMyr," contradicting their prior statement about low background. Please clarify in the text.

2) What is the evidence that "protein tagging in vivo was non-toxic"? It seems that the statement should be limited to the 16h incubation used unless long-term culture experiments in the presence of YnMyr were performed. It is also unclear how the extracellular labeling in Figure 1—figure supplement 1B was performed. Please clarify in the text.

3) In Figure 1B, it is formally possible that alkaline treatment is altering proteins beyond the cleavage of the GPI-anchored label (e.g. causing precipitation or aggregation). The authors should repeat the alkaline treatment after blotting onto the membrane, prior to detection. Alternatively, they could include the noted caveat in the text.

4) In Figure 2D, could the authors explain the YnMyr/Myr-enriched proteins that are categorized as "Non-targets"? Were they only identified in a single experiment? Which experiment are fold enrichment and statistics for this figure drawn from?

5) Can the authors address the different patterns of labeling observed in Figure 1—figure supplement 1A-C compared to Figure 2—figure supplement 2E? Please provide an explanation in the text.

6) It is unclear why there was decreased background in TEV-I compared TEV-II, when in both cases the analyte is TEV-eluted and should therefore experience the same background reduction compared to reagent 2. Please clarify in the text.

7) At several points, the authors compare their results to those of the previous palmitoylation studies to identify myristoylated proteins that were not palmitoylated. It would be informative to perform the converse analysis, to identify palmitoylated proteins that were not identified as myristoylated. Such an analysis might be an orthogonal way of identifying proteins that might have been missed.

8) For Figure 6F-G, since invasion and attachment are performed following rapamycin treatment of all strains, it would be important to include a wildtype strain as the appropriate strain for normalization, instead of normalizing to the complemented strain (i.e. if the complementation is not working properly, we would not be able to tell from Figure 6).

9) If MIC7 is dispensable, were the authors able to isolate knockout parasites from the population and maintain them in culture?

[Editors’ note: further revisions were suggested prior to acceptance, as described below.]

Thank you for submitting your article "Profiling of myristoylation in *Toxoplasma gondii* reveals an N-myristoylated protein important for host cell penetration" for consideration by *eLife*. Your article has been reviewed by Dominique Soldati-Favre as the Senior Editor, a Reviewing Editor, and three reviewers. The reviewers have opted to remain anonymous.

The reviewers have discussed the reviews with one another and the Reviewing Editor has drafted this decision to help you prepare a revised submission.

Importantly, the well-performed myristylome is viewed as important and reliable resource for the scientific community whereas the work on the biology of MIC7 substrate is robust and intriguing but still full of enigma that cannot be addressed in given timeframe.

In consequence we encourage you to consider submitting a revision of your manuscript as a Resource with the MIC7 section serving as atypical substrate validation. Since no new experiments are requested the lockdown situation should not present a barrier.

Summary:

Myristoylation is an irreversible lipid modification that is important for priming subsequent acylation, but on its own has been shown to have additional regulatory roles including facilitating protein-protein interactions. This modification is catalyzed in the cytoplasm by N-myristoyl transferase (NMT). In the protozoan parasite, *Toxoplasma gondii*, the functional significance of myristoylation has been studied for a handful of proteins. Broncel et al., use chemoproteomic methods to globally identify myristoylated proteins-including many modified peptides-in *Toxoplasma*. Using these approaches, the authors identify known myristoylated proteins as well as novel candidates and choose to pursue one of these, MIC7, for additional characterization.

Essential revisions:

1) For the pilot experiment, to define a protein as myristoylated, the authors required robust enrichment (YnMyr/Myr log_2_ fold change > 2), the presence of an MG motif, and insensitivity towards base treatment. It would be helpful to assess this enrichment cutoff by plotting the enrichments of all proteins and highlight known gold standards.

2) The authors claim that there are no substantial differences between the YnMyr and Myr proteomes by comparing the protein abundance of supernatants. Given the near-complete pulldown efficiency (Figure 1—figure supplement 1C), no conclusions should be made about overall substrate abundance from the supernatant.

3) The banding observed in Figure 2—figure supplement 2E doesn't seem to correspond to bands observed in Figure 1—figure supplement 1A and Figure 1—figure supplement 1B. Any background due to endogenously biotinylated proteins (for Figure 2—figure supplement 2E) should be prevalent in the left three lanes.

4) In the text, the authors suggest that the overlap in the number of myristoylated peptides is due to different specificities. Is there any reason to believe that the two cleavable reagents should have different specificities? Another explanation is just experimental variation: for trypsin, of 24 peptides identified, 7 were identified in all three replicates, while 16 were identified in two replicates.

5) In describing the results from the NMT inhibition experiments, the authors state "14 proteins for which the myristoylated peptide was detected were either not quantified at the highest concentration of IMP-1002 or did not respond robustly to NMT inhibition." Lack of detection and lack of robust response are quite different, and the text should reflect this. For example, NMT inhibition could result in protein substrate instability-in fact, the majority of this subset are not represented in the total protein abundance (Toxo proteome tab of Supplementary file 3). As a minor point, by my count in Supplementary file 3, this number is 15, not 14.

6) The authors should compare their findings (65 predicted proteins) with the previously published in silico prediction of myristoylated proteins, 157 proteins as shown below (Alonso et al., 2019. Exploring protein myristoylation in *Toxoplasma gondii*).

7) The co-IPs of MIC7 in Figure 7B are performed with low amounts of detergent (0.2% Triton X-100), which may not fully solubilize micronemes. In order to make any claims about this interaction, some controls should be included in the western blot (e.g., MIC2) to show the specificity. However, the evidence that the myristoylation is not required for localization is sufficiently shown by localization in RAPA treated parasites (Figure 7 figure supplement 1H). It might therefore be prudent for authors to include this caveat in the text, because the evidence presented is not sufficient to demonstrate dimerization.

8) The fact that the myristoylation affects significantly invasion but not parasite attachment (Figure 6F and G) suggests a could be possible a role of MIC7 myristoylation in a signaling cascade leading to rhoptry discharge? The authors should discuss it and also in the context of MIC8 previous implication in this event.

---

## [Author Response]

[Editors’ note: the authors resubmitted a revised version of the paper for consideration. What follows is the authors’ response to the first round of review.]

Reviewer #1:The paper by Broncel et al., reports the identification by using targeted MS, of 65 Myristoyled proteins in Toxoplasma gondii, including CDPK1 and MIC7 (a predicted type-I-transmenbrane protein. The authors provided data, suggesting that Myristoylation of MIC7 is important in host cell invasion.This is a very interesting paper in the field of lipid modifications in the obligate protozoan parasite Toxoplasma gondii and I particularly appreciated the strategy(ies) used to depict these tricky modifications on proteins. Nonetheless, the specificity of the MIC7Myristoylation effect claimed in the paper, which is not given a mechanistic explanation except for the possible involvement in host cell invasion, should be thoroughly re-considered to make the paper really convincing.The experiments appear accurately performed and well presented but a number of points required clarification.Major concerns:1) As the authors reported and already known, YnMyr can be non-specifically incorporated on amino acids differently than N-terminal glycines given the high level of non-myristoylation dependent background. To identify the YnMyr incorporation into N-terminal glycines, the commonly accepted method is the use of NMT inhibitors.I understand the authors' method to distinguish between incorporation of YnMyr in N-Gly or GPI anchors, but to be convincing and to validate the base treatment the authors should also show data with NMT inhibitors for recovery of true Myred proteins.

Following reviewer’s suggestion, we performed additional validation using an NMT inhibitor. In the absence of dedicated *Tg*NMT inhibitors, we have used a previously published *Pf*NMT inhibitor IMP1002 (Schlott et al., 2019). Based on the structural identity (~ 60%) of both *Toxoplasma* and *Plasmodium* NMTs we predicted that this inhibitor will act also on *Tg*NMT. This hypothesis was supported by homology modelling of *Tg*NMT onto *Pv*NMT crystal structure with the inhibitor bound, which showed that all residues involved in compound binding are located in conserved positions in the active site. Following preliminary experiments testing dose response for IMP-1002 in *Toxoplasma*, we performed a large scale MS-based experiment that allowed for quantification of substrate response to NMTi. Importantly, all proteins previously assigned as base sensitive showed no response to inhibitor thus validating the base treatment as means to distinguish YnMyr incorporation at the N-terminal glycine vs GPI anchors. All the described data have been added to the manuscript (see also Figure 3, Figure 3—figure supplement 1 and Supplementary file 3).

2) Authors should report the list of 206 human proteins claimed to bear the MG myristoylation motif (subsection “Identification of the myristoylated proteome in T. gondii”) and compare the retrieved proteins with the recent reported complete list of human Myred proteins (Castrec et al., 2018) to validate their approach. This list is missing for the moment.

We originally did not include the human targets into our manuscript because many were already published elsewhere and the manuscript focuses on the substrates of the parasite NMT and their potential functions. However, after the reviewer’s suggestion, we now included the list of all identified human proteins bearing the MG motif and also compared it with the suggested literature (see manuscript text and Supplementary file 2).

3) Alternative experiments concerning the cellular localization of CDPK1 will help better to understand the function of lipidation of this protein in Toxoplasma gondii. The data presented are not convincing.

After careful consideration of the reviews, we decided to fully focus on the unusual myristoylation of MIC7, rather than putting more emphasis on CDPK1. The CDPK1 data will now be incorporated into a different manuscript that specifically focuses on CDPK1 function and myristate-dependent target activity.

4) As reported above, no mechanistic inside is reported to explain how myristoylation of MIC7 plays a role in the invasion of the host cells. This is a very important aspect of the work that should be further investigated.

We put significant effort into further investigation of the function of MIC7 and its myristoylation. In order to investigate the fate of MIC7 N-terminus and the myristate itself, we generated new complemented WT and Mut lines with Myc tag inserted within the MIC7 EGF2 domain in addition to the Ty1 tag that has previously been present at the C-terminus. After thorough validation (see text, Figure 7 and Figure 7—figure supplement 1) we used these new tools to prove that although not important for MIC7 sorting, the myristate is indeed retained in the lumenal part of the microneme, suggesting that it may function later in the lytic cycle. Consistent with this hypothesis, using large scale invasion assays as well as static and live microscopy, we confirm that both MIC7 and its myristoylation are important in the host cell invasion, particularly in its onset (see Figure 7, Video1, Video 2, Video 3, Video 4)). We then investigate in more detail potential causes of the observed invasion defect and show that it is not due to the faulty microneme secretion, gliding motility or the attempt of tachyzoites to invade, but a likely failure to properly engage with the host cell (Figure 7 and Figure 7—figure supplement 2). We also show that MIC7 behaves unlike most other microneme proteins as it is not shed during invasion. We have tentative data that rhoptry secretion *per se* is not affected, but observe a reduction in cells into which ROP16 has been efficiently transferred if MIC7 is deleted (Figure 7J). Further work will be required to understand the details of the invasion defect. This, we believe, will require substantial work, especially to understand the function of the myristate, whether it may be inserted into the host cell membrane or is required for protein-protein interactions. In IPs using the tagged MIC7 lines, we did not reliably identify specific interaction partners, despite efficient pull down of MIC7 (this data is not shown or discussed in the manuscript). We therefore believe that any MIC7 interaction with partner proteins may be of lower affinity, again, requiring substantial optimisation of IP conditions and will be focus of future work.

5) Of all myristoylated proteins identified is not clear why the authors decided to focus on the two specific proteins (CDPK1 and MIC7). Although I can understand the characterization of MIC7, I'm a bit sceptic about the advances brought by CDPK1 data (already known to undergo myristoylation in most organism).

The CDPK1 dataset has been removed from the manuscript and we now fully focus on MIC7, which, as a secreted transmembrane protein is highly unusual to be myristoylated and the first example of any protein without a signal peptide (to our knowledge) that is trafficked into the micronemes in either *Toxoplasma* or related parasites.

A note to the reviewer: We agree that many CDPKs may be myristoylated in other organisms. But it is important to note that one cannot infer that CDPK1 of *Toxoplasma* is the functional orthologue of CDPK1 in *Arabidopsis*, or any other organism as the CDPK1 indexing has been predominantly defined by the sequence of their discovery, not sequence identity to other species.

Reviewer #2:Broncel et al., uses chemoproteomic techniques to identify myristoylated proteins in *Toxoplasma* and then validate two targets and derive functional information about the role of this modification during the lytic cycle. Myristoylation is a non-reversible modification (as compared to palmitoylation) and can occur either by itself or in partnership with nearby palmitoylation. Indeed, it is dogma that myristoylation is required for subsequent palmitoylation, at least in *Toxoplasma*. This work proceeds work done on the identification of palmitoylated proteins in both *Toxoplasma* and Plasmodium as well as the identification of myristoylated proteins in *P. falciparum* and investigations on its importance. Further, proteins that are both myristoylated and palmitoylated have been predicted, their localisation and function of these sites, in combination, investigated. The substrates that Broncel et al., identify here include these dually acylated proteins, validating their approach and also include many novel molecules. Broncel also duly acknowledge all this previous work.Whilst this reviewer is not a chemical biologist, the approach taken to identify myristoylated proteins seems robust, well controlled and has been well validated. This dataset will be useful for others that are interested in acylation in Toxoplasma or determining whether their protein of interest is myristoylated. Further, this dataset also provides potential information on whether Toxoplasma proteins have a GPI anchor.Presumably, because of the extensive work been done on dual acylation (myristoylation and palmitoylation together) the authors chose to study proteins that they suspect (but not proven) to be only myristoylated and not palmitoylated. They were able to show that the myristoylation site of both CDPK1 and MIC7 are functionally important, albeit limited.The main issues I have with this paper is one of novelty. Whilst this work is robust, well validated and its functional relevance investigated, none of this is surprising. The real interesting question is, and where the field is at, is understanding what myristoylation is doing over and above membrane affinity and what relevance does this have to parasite biology? Again, it is unsurprising that loss of myristoylation leads to changes in solubility (as a surrogate for localisation), but the real interest is what other function might this modification have? This could be particularly interesting with MIC7 where it appears not to have an SP, and myristoylation not required for apparent correct localisation. What about is whether myristoylation is required for surface display or secretion? This might review novel biology of this lipid modification.

See comment to reviewer 1, point 4.

Reviewer #3:Acylation has been shown to influence eukaryotic protein function in a variety of contexts. In Toxoplasma gondii several studies have shown acylation to be important for specific proteins, and global studies have been undertaken to describe the complement of palmitoylated proteins. However, the full extent of myristoylation-a modification often required to license palmitoylation, but also important in its own right-has not been systematically examined in the context of the T. gondii proteome. The present study is an important resource, addressing this gap in our understanding. Through several complementary mass-spectrometry experiments, the authors identify novel and known myristoylated proteins. For two of the identified proteins, CDPK1 and MIC7, the authors use complementation of knockouts to demonstrate the functional importance of myristoylation. Although, the precise role of myristoylation is not uncovered for either candidate, the evidence supporting a role for the modification is strong. The major concerns listed below therefore fall into two main categories: (1) curating the results and interpretation of the myristoylome to ensure its utility as a resource, and (2) ensuring the rigor of test cases.

General comment re. curation and interpretation of the myristoylome:

All proteomic data will be submitted to ProteomeXchange consortium and the target list incorporated within ToxoDB resource to allow broad application of the generated dataset.

1) In reference to Figure 1—figure supplement 1A, the statement "without any detectable background" is not supported by the visible background in the figure. The authors later also state that "[t]his additional layer of confidence in target identification is especially important given the high level of non-myristoylation dependent background reported for metabolic tagging with YnMyr," contradicting their prior statement about low background. Please clarify in the text.

We meant that when applying in-gel fluorescence based detection, there is usually very little background detected (especially seen in Figure 1—figure supplement 1C); whereas when analysed by MS, which is much more sensitive, the level of detected background is quite substantial.

This has been clarified in the text.

2) What is the evidence that "protein tagging in vivo was non-toxic"? It seems that the statement should be limited to the 16h incubation used unless long-term culture experiments in the presence of YnMyr were performed. It is also unclear how the extracellular labeling in Figure 1—figure supplement 1B was performed. Please clarify in the text.

Yes, the non-toxicity of labelling is limited to 16 h of experiment and it has been clarified in the text.

The labelling step for intracellular and extracellular parasites is the same. The term extracellular was used to reflect that the parasites were lysed out of host cells prior to click chemistry and visualization. In contrast, the intracellular means that intracellular parasites were processed (parasite + host cells). Both terms are explained in the figure caption.

3) In Figure 1B, it is formally possible that alkaline treatment is altering proteins beyond the cleavage of the GPI-anchored label (e.g. causing precipitation or aggregation). The authors should repeat the alkaline treatment after blotting onto the membrane, prior to detection. Alternatively, they could include the noted caveat in the text.

This is an important point. We believe we control for this by measuring global protein levels. If some proteins were to precipitate or aggregate, and therefore be reduced in amounts, this would be reflected in the global protein abundance (Supplementary file 1), which we did not observe.

4) In Figure 2D, could the authors explain the YnMyr/Myr-enriched proteins that are categorized as "Non-targets"? Were they only identified in a single experiment? Which experiment are fold enrichment and statistics for this figure drawn from?

The term ‘Non-targets’ was used to represent all proteins that were not classified as targets, i.e. non MG proteins that are non-specifically bound to the beads (background), non-MG proteins that are YnMyr-enriched but not via the N-terminal glycine, e.g. GPI anchored proteins, as well as proteins identified with only one of three capture reagents. We have now updated the legend in the figure and included this extended explanation in the figure caption.

The data presented in Figure 2D were taken from the experiment with the Trypsin cleavable reagent for which we used a t-test to obtain statistical significance (n=3) and also obtained most comprehensive target coverage. This information was added to the figure caption.

5) Can the authors address the different patterns of labeling observed in Figure 1—figure supplement 1A-C compared to Figure 2—figure supplement 2E? Please provide an explanation in the text.

The patterns of labelling look slightly different as two different methods were used for visualization, i.e. in-gel fluorescence and Western blotting with streptavidin HRP. They differ in their sensitivity.

The visualization methods used are clearly indicated.

6) It is unclear why there was decreased background in TEV-I compared TEV-II, when in both cases the analyte is TEV-eluted and should therefore experience the same background reduction compared to reagent 2. Please clarify in the text.

TEV I and TEV II are two alternative strategies that can be applied when using reagent 3 (as shown in Figure 2A). They both use TEV and trypsin for protein elution/digestion, however they differ in the applied enzyme order. TEV I strategy which releases both unmodified and myristoylated peptides in one pool (a result of sequential TEV and trypsin digest) should decrease the level of background. This is due to the fact that TEV protease is used first and will only cleave metabolically tagged proteins that have been clicked to capture reagent 3 (that is equipped with a TEV linker, see Figure 2—figure supplement 2A) and not the ones which are non-specifically bound to the beads. In case of TEV II strategy, trypsin is applied first and all unmodified peptides are collected in fraction1. These peptides will originate from both specific enrichment as well as non-specific binding, hence no reduction in background. Subsequently TEV protease is applied and all myristoylated peptides are now cleaved and collected in a separate fraction, which due to reduced complexity should increase the myristoylated peptide discovery by MS.

For better clarity additional explanation was included in the manuscript.

7) At several points, the authors compare their results to those of the previous palmitoylation studies to identify myristoylated proteins that were not palmitoylated. It would be informative to perform the converse analysis, to identify palmitoylated proteins that were not identified as myristoylated. Such an analysis might be an orthogonal way of identifying proteins that might have been missed.

This is a good idea and we have performed the requested analysis. We found 6 palmitoylated proteins with the potential of being myristoylated, i.e. containing the MG motif. Only one of them has been found in our global analysis and was classified as base sensitive. We did not obtain any experimental evidence for the remaining 5 proteins. As the presence of the MG motif is merely a prerequisite and not a proof of myristoylation, we did not add these proteins to our substrate list.

8) For Figure 6F-G, since invasion and attachment are performed following rapamycin treatment of all strains, it would be important to include a wildtype strain as the appropriate strain for normalization, instead of normalizing to the complemented strain (i.e. if the complementation is not working properly, we would not be able to tell from Figure 6).

The reviewer is correct that the epitope tagged MIC7 version could have a slight growth defect (as indeed observed (Figure 7C) for the new ^Myc^cWT MIC7 with the Myc-tag in the ectodomain).

However, in plaque assays with the old complemented line we did not observe obvious differences in plaque size (see Figure 6E), suggesting that the complementation is working close to or at normal levels. We have not included the parental control as the purpose of the experiment was to investigate the function of myristoylation, which needs to be done in the isogenic iKO background. However, when analysing invasiveness of the new Myc tagged complemented lines (due to the small growth defect) we did include this important control (see Figure 7E).

9) If MIC7 is dispensable, were the authors able to isolate knockout parasites from the population and maintain them in culture?

Yes, after several weeks we were able to isolate clones of MIC7 KO parasites. The clones initially grew very slow, supporting our initial plaque assay results. In addition, we frequently observed many free floating parasites and very few intracellular ones, supporting the observed defect in invasion. After ~3 weeks in culture the KO parasites acquired normal growth kinetics. We hypothesized that a bradyzoite MIC7-like protein (TGME49_315520) identified by blast analysis could have been upregulated by the KO parasites as a potential compensation mechanism. This turned out not to be the case, as verified by cDNA analysis from these parasites (data not shown). We have decided not to include this information within the manuscript as for now it is mainly based on observations, rather than solid measurements, and will require more work, including RNA seq analysis to identify potentially compensating transcripts, their validation and functional analysis and is, we think, beyond the scope of this manuscript.

[Editors’ note: what follows is the authors’ response to the second round of review.]

Importantly, the well performed myristylome is viewed as important and reliable resource for the scientific community whereas the work on the biology of MIC7 substrate is robust and intriguing but still full of enigma that cannot be addressed in given timeframe.In consequence we encourage you to consider submitting a revision of your manuscript as a Resource with the MIC7 section serving as atypical substrate validation. Since no new experiments are requested the lockdown situation should not present a barrier.Summary:Myristoylation is an irreversible lipid modification that is important for priming subsequent acylation, but on its own has been shown to have additional regulatory roles including facilitating protein-protein interactions. This modification is catalyzed in the cytoplasm by N-myristoyl transferase (NMT). In the protozoan parasite, Toxoplasma gondii, the functional significance of myristoylation has been studied for a handful of proteins. Broncel et al., use chemoproteomic methods to globally identify myristoylated proteins-including many modified peptides-in Toxoplasma. Using these approaches, the authors identify known myristoylated proteins as well as novel candidates and choose to pursue one of these, MIC7, for additional characterization.Essential revisions:1) For the pilot experiment, to define a protein as myristoylated, the authors required robust enrichment (YnMyr/Myr log_2_ fold change > 2), the presence of an MG motif, and insensitivity towards base treatment. It would be helpful to assess this enrichment cutoff by plotting the enrichments of all proteins and highlight known gold standards.

The requested assessment and the corresponding plot have been made and included in the revised manuscript (see Results section and Figure 2—figure supplement 2C).

2) The authors claim that there are no substantial differences between the YnMyr and Myr proteomes by comparing the protein abundance of supernatants. Given the near-complete pulldown efficiency (Figure 1—figure supplement 1C), no conclusions should be made about overall substrate abundance from the supernatant.

We used post enrichment supernatants to test for general proteome changes that would have been indicative of adverse effects invoked by YnMyr treatment and potentially affect our conclusions, not to quantify the myristoylated proteins. Comprehensive quantification of myristoylated proteins without prior enrichment would likely not have been possible without extensive sample fractionation or targeted mass spectrometry approaches. We have amended this statement for clarity (see Results section).

3) The banding observed in Figure 2—figure supplement 2E doesn't seem to correspond to bands observed in Figure 1—figure supplement 1A and Figure 1—figure supplement 1B. Any background due to endogenously biotinylated proteins (for Figure 2—figure supplement 2E) should be prevalent in the left three lanes.

The reviewer is correct that any background originating from the endogenous biotinylation should also be present in (-) YnMyr controls in Figure 2—figure supplement 2E . We therefore do not think that these bands correspond to background. We think that it is the difference in the applied visualization methodology (in gel fluorescence (igFL) vs Western blotting (WB)) that causes this inconsistency. Specifically, we believe that the efficiency of protein transfer onto the nitrocellulose membrane prior to incubation with streptavidin HRP may be the cause. Here we applied a semi dry (SD) blotting system which is known to provide faster and simpler transfer than a standard tank transfer method. Unfortunately, due to high electrical field strengths and reduced volumes of transfer buffer, SD systems can cause a blow-through of low molecular weight proteins and incomplete transfer of high molecular weight proteins. We would like to stress that bands from the WB (75 kD, 50-60 kD, 20-25 kD) are also visible by igFL (Figure 1—figure supplement 1A and Figure 1—figure supplement 1B) although with different intensity.

We decided to not pursue this further, as the goal of this experiment was to compare YnMyr enrichment between capture reagents and not to quantify specific bands. We believe that we clearly show in Figure 2—figure supplement 2E that all three capture reagents used enable efficient pull down of myristoylated proteins, which we then quantify using mass-spectrometry.

4) In the text, the authors suggest that the overlap in the number of myristoylated peptides is due to different specificities. Is there any reason to believe that the two cleavable reagents should have different specificities? Another explanation is just experimental variation: for trypsin, of 24 peptides identified, 7 were identified in all three replicates, while 16 were identified in two replicates.

The two cleavable reagents (Trypsin and TEV) used in our study are very different in terms of their structure, applied click reaction conditions and cleavage strategy, and finally in the nature of the resulting myristoylation adduct that is detected by MS. We were referring to all these features as to ‘different specificities’. We realize, however, that such statement was misleading for a reader in suggesting that different subsets of substrates are being targeted by these two reagents. We have amended the text accordingly (see Results section).

5) In describing the results from the NMT inhibition experiments, the authors state "14 proteins for which the myristoylated peptide was detected were either not quantified at the highest concentration of IMP-1002 or did not respond robustly to NMT inhibition." Lack of detection and lack of robust response are quite different, and the text should reflect this. For example, NMT inhibition could result in protein substrate instability-in fact, the majority of this subset are not represented in the total protein abundance (Toxo proteome tab of Supplementary file 3). As a minor point, by my count in Supplementary file 3, this number is 15, not 14.

We recognize how this statement could have been misleading and have amended the text accordingly (see Results section). Regarding the Reviewer’s minor point addressing the number of proteins with myristoylated peptide, the reported number (14) is actually correct, the mistake was that one of the protein entries (289615) had been entered twice. We thank for that comment and we amended the table accordingly.

6) The authors should compare their findings (65 predicted proteins) with the previously published in silico prediction of myristoylated proteins, 157 proteins as shown below (Alonso et al., 2019. Exploring protein myristoylation in Toxoplasma gondii).

The requested comparison has been made and included in the revised manuscript (see Results section and Supplementary file 4).

7) The co-IPs of MIC7 in Figure 7B are performed with low amounts of detergent (0.2% Triton X-100), which may not fully solubilize micronemes. In order to make any claims about this interaction, some controls should be included in the western blot (e.g., MIC2) to show the specificity. However, the evidence that the myristoylation is not required for localization is sufficiently shown by localization in RAPA treated parasites (Figure 7 - figure supplement 1H). It might therefore be prudent for authors to include this caveat in the text, because the evidence presented is not sufficient to demonstrate dimerization.

This was a good suggestion and could well have been the case. Using the previously prepared lysates we have included the MIC2 control, see Figure 7B in the revised manuscript. This shows that MIC2 is not co-immunoprecipitated in the MIC7 IPs, indicating efficient lysis of the micronemes and dimerization or oligomerisation of MIC7, under the conditions used.

8) The fact that the myristoylation affects significantly invasion but not parasite attachment (Figure 6F and G) suggests a could be possible a role of MIC7 myristoylation in a signaling cascade leading to rhoptry discharge? The authors should discuss it and also in the context of MIC8 previous implication in this event.

Indeed, we were tempted to speculate that MIC7 may be involved in such a cascade, similarly to MIC8. However, since in case of MIC7 the rhoptry secretion phenotype was not as severe as for MIC8, we believe that rhoptries are likely to discharge but the contents are not efficiently translocated into the host cell. We have clarified this and referenced MIC8 in the revised manuscript (see Discussion section).